# Synergy between SIRT1 and SIRT6 helps recognize DNA breaks and potentiates the DNA damage response and repair in humans and mice

Fanbiao Meng[1,2†], Minxian Qian[1,3†], Bin Peng[3†], Linyuan Peng[1,3], Xiaohui Wang[1,4], Kang Zheng[5], Zuojun Liu[1,3], Xiaolong Tang[1,3], Shuju Zhang[3], Shimin Sun[1,5], Xinyue Cao[1,3], Qiuxiang Pang[5], Bosheng Zhao[5], Wenbin Ma[6], Zhou Songyang[6], Bo Xu[2], Wei-Guo Zhu[3,4], Xingzhi Xu[3,4]*, Baohua Liu[1,3,4,7]*

[1]Shenzhen Key Laboratory for Systemic Aging and Intervention, National Engineering Research Center for Biotechnology (Shenzhen), Shenzhen University Health Science Center, Shenzhen, China; [2]The Key Laboratory of Breast Cancer Prevention and Therapy, Ministry of Education, Tianjin's Clinical Research Center for Cancer, Tianjin Medical University Cancer Institute and Hospital, National Clinical Research Center for Cancer, Tianjin, China; [3]Guangdong Key Laboratory of Genome Stability and Human Disease Prevention, School of Basic Medical Sciences, Shenzhen University Health Science Center, Shenzhen, China; [4]International Cancer Center, Shenzhen University Health Science Center, Shenzhen, China; [5]Anti-aging & Regenerative Medicine Research Institution, School of Life Sciences, Shandong University of Technology, Zibo, China; [6]Key Laboratory of Gene Engineering of the Ministry of Education and State Key Laboratory for Biocontrol, School of Life Sciences, Sun Yat-sen University, Guangzhou, China; [7]Guangdong Provincial Key Laboratory of Regional Immunity and Diseases, School of Basic Medical Sciences, Shenzhen University Health Science Center, Shenzhen, China

*For correspondence:
xingzhi.xu@szu.edu.cn (XX);
ppliew@szu.edu.cn (BL)

†These authors contributed equally to this work

Competing interests: The authors declare that no competing interests exist.

**Abstract** The DNA damage response (DDR) is a highly orchestrated process but how double-strand DNA breaks (DSBs) are initially recognized is unclear. Here, we show that polymerized SIRT6 deacetylase recognizes DSBs and potentiates the DDR in human and mouse cells. First, SIRT1 deacetylates SIRT6 at residue K33, which is important for SIRT6 polymerization and mobilization toward DSBs. Then, K33-deacetylated SIRT6 anchors to γH2AX, allowing its retention on and subsequent remodeling of local chromatin. We show that a K33R mutation that mimics hypoacetylated SIRT6 can rescue defective DNA repair as a result of *SIRT1* deficiency in cultured cells. These data highlight the synergistic action between SIRTs in the spatiotemporal regulation of the DDR and DNA repair in humans and mice.

## Introduction

DNA damage can be induced by various endogenous and exogenous agents. Upon detection of damage, the DNA damage response (DDR) is immediately elicited to regain genomic integrity via chromatin remodeling, signaling transduction and amplification (*Ciccia and Elledge, 2010*). Double-strand breaks (DSBs) are the most severe type of DNA lesion; they are recognized by the Mre11-Rad50-Nbs1 (MRN) complex, which recruits and activates phosphatidylinositol 3-kinase-like protein kinase ataxia-telangiectasia mutated (ATM) or ATM- and Rad3-related (ATR). H2AX is then rapidly

phosphorylated (γH2AX) by ATM/ATR, and serves as a platform to localize repair proteins near to the DNA breaks (*Celeste et al., 2003*). Simultaneously, various histone-modifying enzymes, hetero-chromatin factors and ATP-dependent chromatin remodelers work cooperatively to relax the chromatin structure and ensure that additional repair factors have access to the DSBs (*Price and D'Andrea, 2013*). Despite all these advances in understanding the DDR, how DSBs are initially and precisely recognized is largely unknown.

NAD$^+$-dependent sirtuins belong to class III histone deacetylases (HDACs) (*Houtkooper et al., 2012*). Seven sirtuins (SIRT1-7) with various enzymatic activities and physiological functions are expressed in mammals. SIRT1, 6 and 7 localize in the nucleus and seem to be most relevant to the DDR as their depletion causes growth retardation, a defective DDR and DNA repair and premature aging (*Mostoslavsky et al., 2006*; *Wang et al., 2008*; *Vazquez et al., 2016*). Upon DNA damage, SIRT1 redistributes on chromatin, co-localizes with γH2AX, and deacetylates XPA, NBS1 and Ku70 to regulate nucleotide excision repair, homologous recombination (HR) and non-homologous end-joining (NHEJ) (*Fang et al., 2016*; *Yuan et al., 2007*; *Fan and Luo, 2010*; *Jeong et al., 2007*). Depleting *Sirt1* in mouse fibroblasts impairs the DDR and leads to genomic instability (*Wang et al., 2008*). SIRT6 is one of the earliest factors recruited to DSBs; it initiates the subsequent recruitment of SNF2H, H2AX, DNA-PKcs and PARP1 (*Atsumi et al., 2015*; *McCord et al., 2009*; *Van Meter et al., 2016*). SIRT6 mono-ribosylates PARP1 to enhance its activity (*Mao et al., 2011*). Despite their rapid mobilization to DNA breaks, the triggers for sirtuin recruitment are obscure (*Vazquez et al., 2016*; *Dobbin et al., 2013*; *Toiber et al., 2013*).

Here, we aimed to delineate the mechanisms underlying DSB recognition. We found that SIRT6 polymerizes and directly recognizes DSBs via a putative DNA-binding pocket consisting of N- and C-termini from two adjacent molecules. SIRT1 interacts with SIRT6 and deacetylates it at K33, thus allowing its polymerization and recognition of DSBs. A K33R mutant, mimicking hypoacetylated SIRT6, could rescue DNA repair defects in *SIRT1* knockout (KO) cells. Our data highlight an essential synergy between SIRT1 and SIRT6 in the spatiotemporal regulation of the DDR.

## Results

### SIRT6 directly recognizes DNA double-strand breaks

Nuclear SIRTs (SIRT1/6/7) are quickly mobilized to DSBs (*Figure 1—figure supplement 1*) and serve as a scaffold for DNA repair factors (*Vazquez et al., 2016*; *Dobbin et al., 2013*; *Toiber et al., 2013*). Intriguingly, these nuclear SIRTs are also activated by RNA and the nucleosome (*Gil et al., 2013*; *Tong et al., 2017*). We thus reasoned that SIRTs might directly sense DNA breaks, especially DSBs. To test our hypothesis, we established a molecular docking simulation using AutoDock Vina software (*Trott and Olson, 2010*). We obtained the crystal structures for SIRT1 (PDB code 4I5I) (*Zhao et al., 2013*), SIRT6 (PDB code 3PKI) (*Pan et al., 2011*) and SIRT7 (PDB code 5IQZ) (*Priyanka et al., 2016*) from the Protein Data Bank (https://www.rcsb.org). We removed the heteroatoms to expose interaction regions and added Gasteiger charges to construct docking models. A DSB structure was extracted from the PDB code 4DQY (*Langelier et al., 2012*). As SIRTs use NAD$^+$ as a co-substrate in amide bond hydrolysis, which shares a similar skeleton to the phosphate, base and ribose groups on broken DSB ends, we included NAD$^+$ as a simulation control.

We found that the binding affinity between NAD$^+$ and all nuclear SIRTs was within the range of –eight to –10 kcal/mol (*Figure 1A*). Surprisingly, only the binding between the DSB and SIRT6 was energetically favored (–12.7 kcal/mol), being even lower than that of NAD$^+$ (*Figure 1A,B*). This finding suggested that SIRT6 might directly bind to DSBs and prompted us to gain further experimental evidence.

We next generated a DSB-mimicking biotin-conjugated DNA duplex and performed an in vitro pulldown assay. Recombinant SIRT6 (rSIRT6), but not rSIRT1 or rSIRT7, bound to the DNA duplex (*Figure 1C*). This finding was verified by a fluorescence polarization (FP) assay using a FAM-labeled DNA duplex. We observed dynamic FP (Kd = 166.3 nM; *Figure 1D*), supporting a specific and direct interaction between the DNA duplex and rSIRT6. By contrast, the FP was minimal for rSIRT1, rSIRT7 and GST control (*Figure 1D*). To interrogate whether such binding is specific to broken DNA, we repeated the pulldown assay in the presence of unlabeled linear or circular DNA. While linearized DNA inhibited rSIRT6 binding to the DNA duplex, circular DNA had a minimal effect (*Figure 1E*).

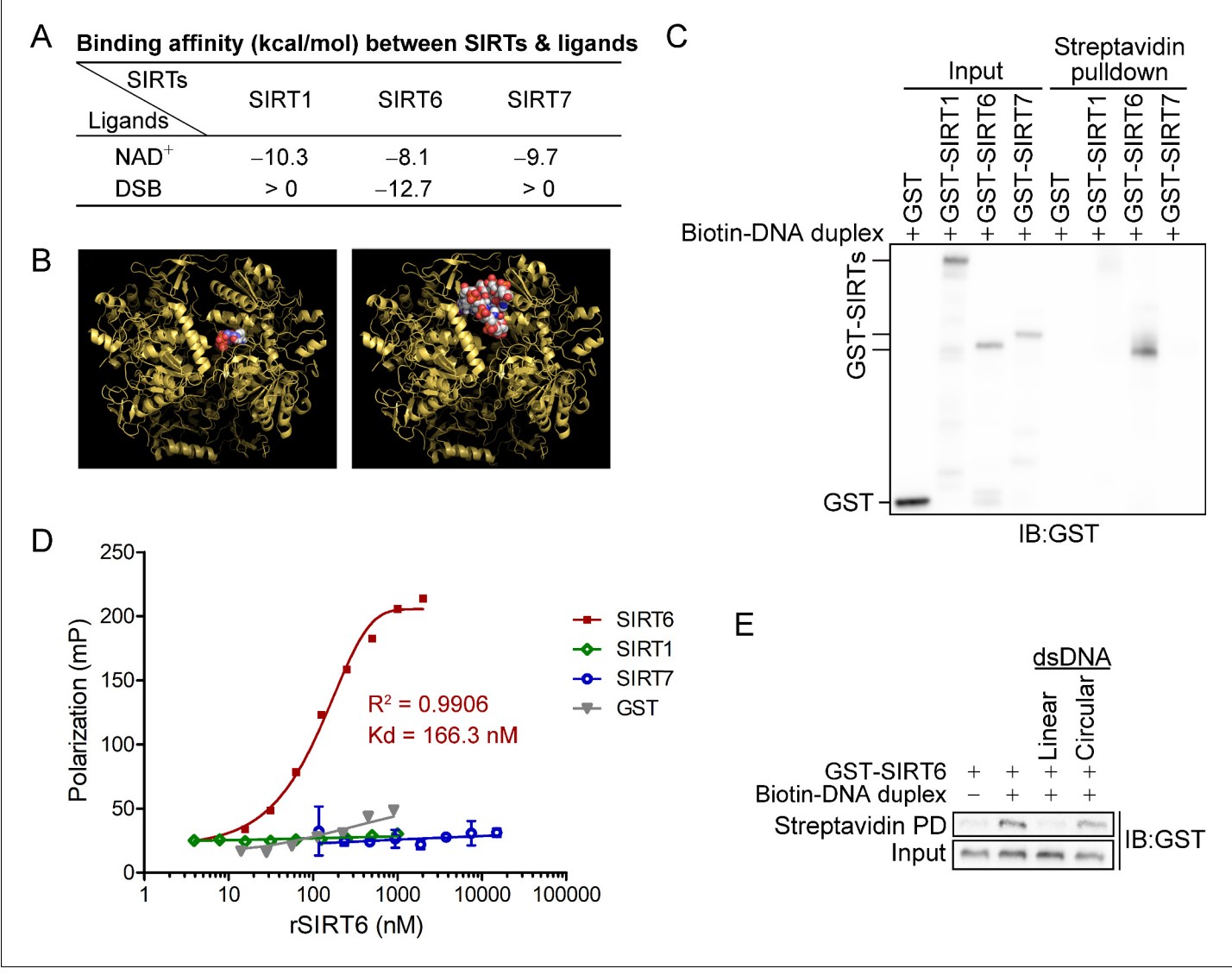

**Figure 1.** SIRT6 directly recognizes DNA breaks. (**A**) The predicted binding affinity (kcal/mol) between sirtuins (SIRTs) and ligands (NAD[+] and a DSB) by molecular docking analysis. (**B**) Molecular docking of SIRT6 with a DSB (right) and NAD[+] (left) using AutoDock Vina software. (**C**) A biotin-labeled DNA duplex was incubated with the indicated recombinant SIRTs. Streptavidin beads were pulled down and blotted with anti-GST antibodies. (**D**) The fluorescence polarization (FP) of FAM-labeled DNA (20 nM) was detected after incubation with GST-SIRT1, GST-SIRT6, GST-SIRT7 or GST at the indicated concentration. (**E**) A pulldown assay comprising a biotin-labeled DNA duplex with GST-SIRT6 in the presence of unlabeled linear DNA or circular DNA.

The online version of this article includes the following figure supplement(s) for figure 1:

**Figure supplement 1.** DSB-recruitment kinetics of SIRTs.

Together, these data indicate that SIRT6, but not SIRT1 or SIRT7 recognizes and directly binds to DSBs.

## SIRT6 is dynamically acetylated in the N terminus at K33

As predicted from the crystallographic data, SIRT6s form an asymmetric hexamer (*Pan et al., 2011*) that generates three potential DSB binding pockets; each of these pockets consist of two N-termini and two C-termini from two adjacent molecules (*Figure 2—figure supplement 1A*). Both the N-termini and C-termini are essential for SIRT6 to associate with chromatin (*Tennen et al., 2010*). To gain biochemical evidence for SIRT6 polymerization, we employed a biomolecule fluorescence compensation system (BiFC). In brief, we cloned *SIRT6* cDNA into either the N-terminal or C-terminal of a

yellow fluorescence protein (YFP), herein termed N-SIRT6 and C-SIRT6. The yellow fluorescence was detectable by FACS only when N-SIRT6 directly interacted with C-SIRT6. After co-transfecting these constructs into HEK293 cells, we detected a strong fluorescence signal by FACS in >24% cells (*Figure 2—figure supplement 1B*), suggesting a direct interaction between the two SIRT6 molecules. By contrast, fluorescence signal was rarely detected in BiFC analysis of N-SIRT6 and C-SIRT3 or non-tagged SIRT6 and C-SIRT3 (*Figure 2—figure supplement 1C*). To confirm this SIRT6 polymerization event, we performed co-immunoprecipitation (Co-IP) in HEK293 cells in which we had co-overexpressed FLAG-SIRT6 and HA-SIRT6. Consistently, we detected FLAG-SIRT6 but not FLAG-SIRT3 in the anti-HA-SIRT6 immunoprecipitates (*Figure 2—figure supplement 1D*).

The DSB phosphate backbone is negatively charged. The positive-charged environment of SIRT6 thus favors its binding to DSBs. Indeed, one of our predicted DSB-binding pockets formed by two adjacent molecules in SIRT6 hexamer consisted of six positively charged residues at the edge, namely four arginine (R32/39) and two lysine (K33) residues (*Figure 2—figure supplement 1E*). Acetylation is the most redundant post-translational modification that converts positively charged K to neutral Kac. This property is utilized by proteins with a lysine-rich domain (KRD), for example Histones, Ku70 and p53, for mediating dynamic interactions with proteins harboring an acidic domain like SET (*Wang et al., 2016*). The heterodimerized Ku70 and Ku80 complex directly senses DSBs via a flexible C-termini containing multiple K residues, and regulates NHEJ (*Hu et al., 2012*). We therefore examined whether SIRT6 is (de)acetylated on these K residues thus sensing DSBs. We immunoprecipitated FLAG-SIRT6 with an anti-FLAG antibody and then probed the immunoprecipitate with anti-Kac antibodies. As shown, Kac was detected in the precipitated FLAG-SIRT6 immunocomplex (*Figure 2A*). We then purified FLAG-SIRT6 and performed high-resolution LC-MS/MS to identify which K residues are acetylated (*Supplementary file 1*). We identified K15 and K33 in the N-terminus. To confirm these acetylated K residues, we generated K15R and K33R point mutants, with K17R serving as a negative control. While neither K15R nor K17R affected the FLAG-SIRT6 acetylation level, K33R significantly inhibited it (*Figures 2A* and *Figure 2—figure supplement 2*), supporting that K33 is dynamically (de)acetylated.

## Dynamic SIRT6 K33 (de)acetylation regulates DSB sensing

To understand the function of SIRT6 K33 acetylation, we examined whether it is required for binding to DSBs. We used our K33R mutant and generated a new K33Q mutant to mimic deacetylated or acetylated SIRT6, respectively (*Tang et al., 2017*). We also mutated SIRT6 H133 to a tyrosine residue (H133Y) to blunt SIRT6 enzymatic activity (*Toiber et al., 2013*). K33Q and H133Y binding to the DNA duplex was significantly compromised compared to WT and K33R binding (*Figure 2B*). Consistently, we recorded notable FP for SIRT6 K33R (Kd = 104.9 nM) but little FP for SIRT6 K33Q (*Figure 2C*).

We then monitored GFP-SIRT6 mobility in cells upon receipt of DNA damage. H133 is critical for enriching SIRT6 on chromatin (*Tennen et al., 2010*). We reconstituted GPF-SIRT6 WT, K33Q, K33R and H133Y in *Sirt6*$^{-/-}$ cells and monitored their recruitment to DSBs. While the K33Q and H133Y mutations significantly jeopardized efficient SIRT6 recruitment to DNA breaks, the SIRT6 K33R mutant retained such ability (*Figure 2D,E*). To gain more experimental support, we made use of an inducible DR-GFP reporter system that contains a unique I-*Sce*I cutting site. In presence of triamcinolone acetonide (TA), the I-*Sce*I-GR enzyme translocated to the nucleus within 10 min and generated DSBs, as evidenced by an increase in the γH2AX level (*Figure 2F,G*). We thus monitored the occupancy of SIRT6 on chromatin surrounding these induced DSBs, by chromatin immunoprecipitation (ChIP) and quantitative PCR, as previously described (*Soutoglou et al., 2007*). Both the K33Q and H133Y mutations compromised SIRT6 recruitment to the sites of damage, whereas SIRT6 K33R recruitment was comparable to that of SIRT6 WT (*Figure 2H*).

Upon DNA damage, the acetylation levels of H3K9 and H3K56 decline, and after repair, goes back to the original level (*Tjeertes et al., 2009*). H3K9ac and H3K56ac are deacetylating targets of SIRT6, indicating that SIRT6 might contribute to the reduced H3K9ac and H3K56ac levels on the DSB-surrounding chromatin. Indeed, reconstituted SIRT6 WT and K33R downregulated the levels of H3K9ac and H3K56ac in *Sirt6*$^{-/-}$ cells, while K33Q and H133Y failed (*Figure 2I and J* and *Figure 2—figure supplement 3A*). Further, the K33Q and H133Y mutations also affected SNF2H recruitment to DSBs (*Figure 2—figure supplement 3B*), which requires SIRT6 (*Toiber et al., 2013*), but no effect

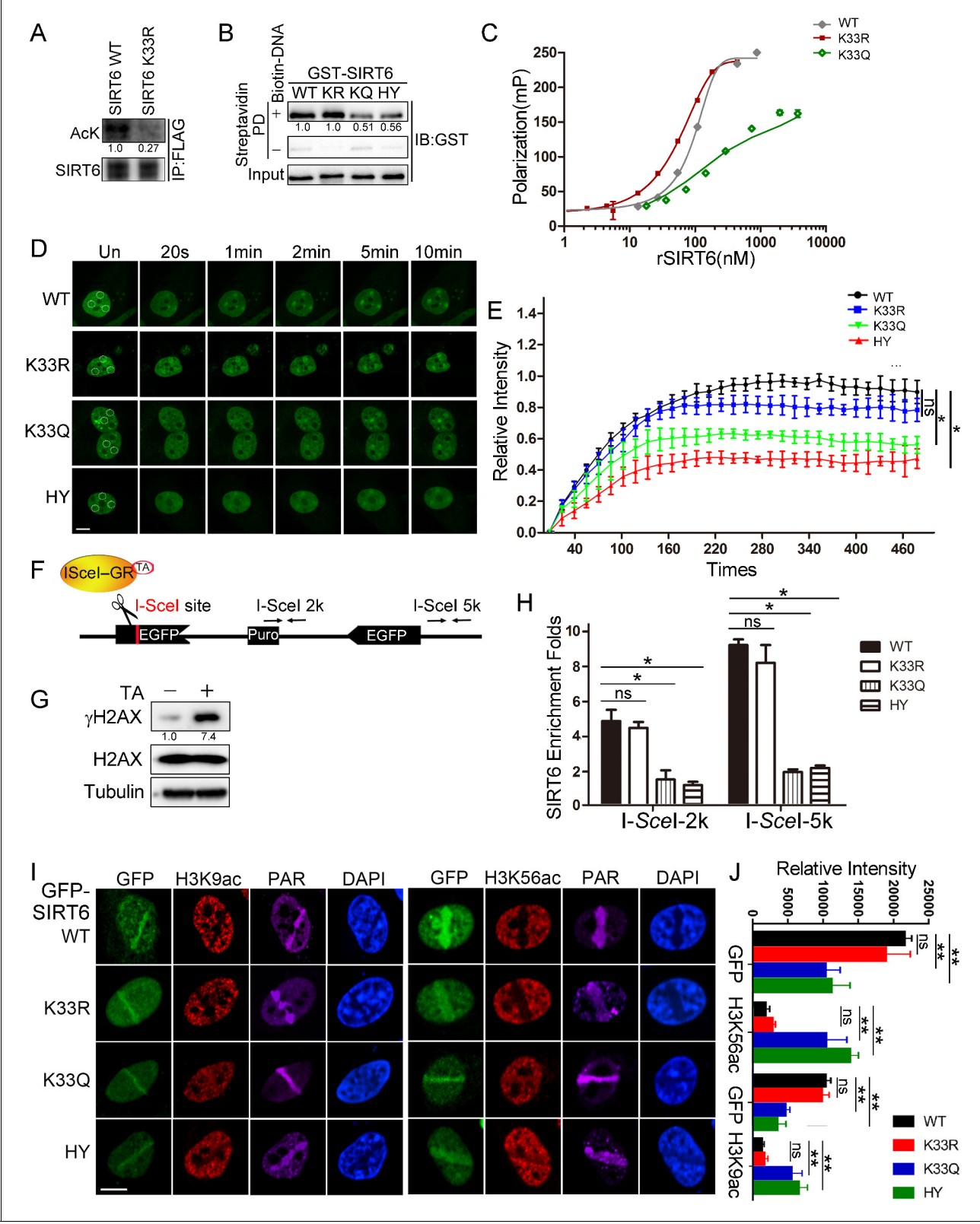

**Figure 2.** SIRT6 K33 (de)acetylation regulates DSB binding. (**A**) The acetylation levels of FLAG-SIRT6 WT and K33R were assessed by western blotting with pan-acetyl antibodies in anti-FLAG immunoprecipitates in HEK293T cells. (**B**) Streptavidin pulldown assay showing the interactions between a biotin-labeled DNA duplex and the indicated GST-SIRT6 recombinant proteins. (**C**) Fluorescence polarization (FP) of Fam-labeled DNA was detected after incubating with GST-SIRT6 WT, K133R or K133Q recombinant proteins. (**D–E**) The dynamic recruitment of GFP-SIRT6, K33R, K33Q and HY (H133Y)

*Figure 2 continued on next page*

*Figure 2 continued*

to laser-induced DNA breaks was assessed by confocal microscopy. Representative images are shown (**D**) and the white dot circles indicate the damage sites. Scale bar, 10 µm. The relative intensity was calculated in Fiji (Image J) (**E**). The data represent the means ± s.e.m., *p<0.05, ns: not significant, n = 30. (**F**) A schematic of the DR-GFP construct, which contains a single I-*Sce*I site to create DNA break in the presence of triamcinolone acetonide and I-*Sce*I endonuclease. The positions of the amplification primers 2K and 5K downstream I-*Sce*I site used for q-PCR are indicated. (**G**) DNA breaks were generated in DR-GFP stably transfected HeLa cells after triamcinolone acetonide (TA) treatment for 20 min, as evidenced by elevated γH2AX staining. (**H**) ChIP-PCR analysis of the enrichment of SIRT6 and various SIRT6 mutants at DNA break sites. The relative SIRT6 expression was confirmed by western blotting. The qPCR data were normalized to the input DNA and a sample not treated with I-*Sce*I endonuclease (no cut). The data represent the means ± s.e.m., *p<0.05, ns: not significant, n = 3. (**I–J**) Fluorescence imaging of GFP-SIRT6 WT, indicated mutants, immune-stained H3K9ac and H3K56ac in *Sirt6*$^{-/-}$ MEFs after laser induced DNA damage. PAR immunostaining reveals the damage site. Scale bar, 10 µm. The relative fluorescence intensity was calculated by Fiji (Image J) (**J**). The data represent the means ± s.e.m., **p<0.01, ns: not significant, n = 30.

The online version of this article includes the following figure supplement(s) for figure 2:

**Figure supplement 1.** Polymerization of SIRT6.
**Figure supplement 2.** Acetylation of SIRT6.
**Figure supplement 3.** Deacetylase activity of SIRT6.
**Figure supplement 4.** Polymerization of SIRT6.

was observed in the presence of the K33R mutation. Of note, neither K33R nor K33Q affected the deacetylase activity of SIRT6 (*Figure 2—figure supplement 3C*).

We next analyzed whether dynamic K33 (de)acetylation modulates SIRT6 polymerization. We co-overexpressed HA-SIRT6 and various FLAG-SIRT6 mutants and performed Co-IP. We detected FLAG-SIRT6 in the anti-HA immunoprecipitates, supporting that SIRT6 polymerization occurs (*Figure 2—figure supplement 1D*). While HA-SIRT6 was still able to bind to FLAG-SIRT6 K33R, its binding to SIRT6 K33Q was significantly jeopardized. Of note, the enzyme-dead H133Y mutation also jeopardized SIRT6 polymerization. This finding is consistent with the fact that the H133 site is important for both SIRT6 deacetylase activity and for mediating the chromatin association (*Tennen et al., 2010*). We confirmed this jeopardized polymerization in the K33Q mutant condition by BiFC assay (*Figure 2—figure supplement 4A,B*). Together, these data implicate that dynamic SIRT6 K33 (de)acetylation modulates SIRT6 polymerization and thus DSB binding.

## SIRT6 interacts with SIRT1

Having confirmed SIRT6 (de)acetylation, we moved to examine potential deacetylase of SIRT6. To this end, we first tested the effect of various HDAC inhibitors on SIRT6 acetylation level. We noticed that the level of acetylated SIRT6 was largely elevated in the presence of the class III HDAC (SIRTs) inhibitor nicotinamide (NAM) or the SIRT1-specific inhibitor Ex527, but not the class I/II HADC inhibitor Trichostatin A (TSA) (*Figure 3—figure supplement 1*). This finding suggested that SIRT1 might be involved in SIRT6 deacetylation. Indeed, co-IP and western blotting revealed that FLAG-SIRT6 interacted with endogenous SIRT1 (*Figure 3A*) and vice versa in HEK293 cells (*Figure 3B*). In addition, we detected SIRT1 in the anti-SIRT6 immunoprecipitates and *vice versa* (*Figure 3C,D*). A GST pulldown assay confirmed that His-SIRT1 directly interacted with GST-SIRT6 (*Figure 3E*). Further, we observed co-localization between SIRT6 and SIRT1 by confocal microscopy in cells co-transfected with GFP-SIRT6 and DsRed-SIRT1 or in cells co-stained with specific antibodies (*Figure 3F* and *Figure 3—figure supplement 2A*).

SIRTs contain a conserved Sir2 domain and flexible N-termini and C-termini. To locate the exact SIRT6 domains that interact with SIRT1, we deleted the N-terminus and C-terminus, as previously reported (*Tennen et al., 2010*; *Figure 3—figure supplement 2B,C*). Western blotting analysis showed that the interaction between SIRT6 and SIRT1 was lost if the N-terminus or C-terminus of SIRT6 was deleted (*Figure 3G*). As the C-terminus contains the nuclear location signal (*Tennen et al., 2010*), we speculate that its depletion may restrict SIRT6 in cytoplasm, thus preventing the interaction between SIRT1 and SIRT6. Thus, the data indicate that SIRT6 physically interacts with SIRT1, most likely through the N-terminus of SIRT6.

## SIRT1 deacetylates SIRT6 at K33

We next examined whether SIRT1 deacetylates SIRT6 via the direct interaction that we identified above. Overexpression of SIRT1 but not of other sirtuins inhibited FLAG-SIRT6 acetylation

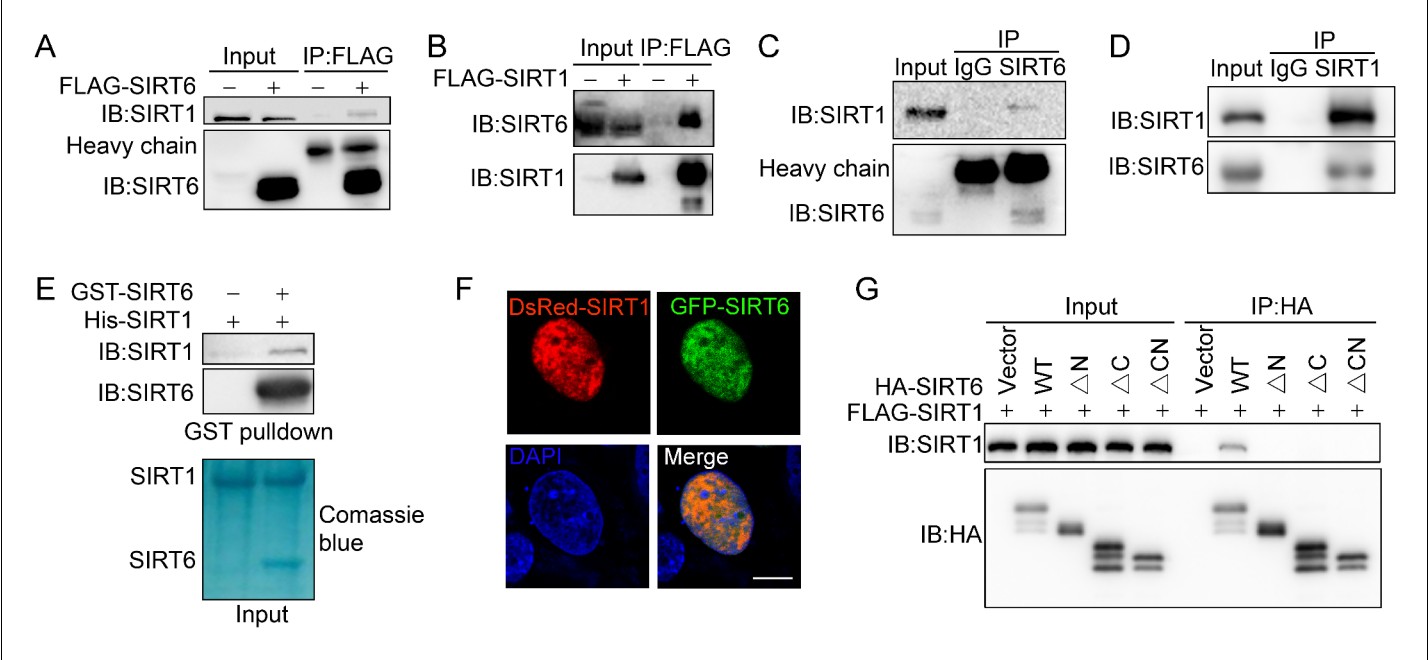

**Figure 3.** SIRT6 interacts with SIRT1. (**A**) Western blot analysis of SIRT1 levels in anti-FLAG immunoprecipitates in HEK293 cells transfected with FLAG-SIRT6 or an empty vector. (**B**) Western blot analysis of SIRT6 levels in anti-FLAG immunoprecipitates in HEK293 cells transfected with FLAG-SIRT1 or an empty vector. (**C**) Western blot analysis of SIRT1 in anti-SIRT6 immunoprecipitates in HeLa cells. (**D**) Western blot analysis of SIRT6 in anti-SIRT1 immunoprecipitates in HeLa cells. (**E**) GST pulldown assay between GST-SIRT6 and His-SIRT1 in vitro. (**F**) Representative images of DsRed-SIRT1 and GFP-SIRT6 localization in U2OS cells, determined by confocal microscopy. Scale bar, 10 μm. (**G**) Co-immunoprecipitation and western blot analysis of FLAG-SIRT1 in HEK293 cells overexpressing FLAG-SIRT1 and HA-SIRT6 ΔN (N-terminus deleted), ΔC (C-terminus deleted) or ΔCN (N-/C-termini deleted).

The online version of this article includes the following figure supplement(s) for figure 3:

**Figure supplement 1.** Acetylation level of SIRT6.

**Figure supplement 2.** SIRT1-SIRT6 interaction.

(*Figure 4A*). Conversely, knocking down *SIRT1* significantly upregulated endogenous SIRT6 acetylation in HEK293 cells (*Figure 4B*). Furthermore, the SIRT6 acetylation level decreased in the presence of ectopic SIRT1 but not in the presence of its catalytic mutant SIRT1-H363Y (*Figure 4C*), despite the two proteins still showing a physical interaction, suggesting that SIRT6 is likely a deacetylation target of SIRT1. To test our hypothesis, we established an in vitro deacetylation assay. We eluted recombinant FLAG-SIRT6 with a FLAG peptide from HEK293 cell lysate. We found that SIRT1 deacetylated SIRT6 in the presence of NAD$^+$, while NAM inhibited this process (*Figure 4D,E*). The deacetylase-inactive SIRT1-H363Y was unable to deacetylate SIRT6.

As SIRT1 might interact with the SIRT6 N-terminus, we hypothesized that it might also deacetylate K33ac. While the acetylation level of SIRT6 was increased in *SIRT1$^{-/-}$* HEK293 cells, that of K33R was hardly affected (*Figure 4F*). Additionally, the acetylation level of SIRT6 K33R was minimally changed upon SIRT1 overexpression (*Figure 4G*), whereas that of K143/145R was downregulated upon ectopic SIRT1 (*Figure 4—figure supplement 1A*). These data support that K33ac is a target of SIRT1. By contrast, the SIRT1 acetylation level was relatively unaffected upon SIRT6 overexpression (*Figure 4—figure supplement 1B*). To further validate these findings, we synthesized a K33ac-containing peptide and found that it effectively blocked the in vitro binding of SIRT6 to SIRT1 (*Figure 4H*). Of note, the GST pulldown assay suggested that the N-terminus rather than the C-terminus of SIRT6 was responsible for its interaction with SIRT1. Together, these data suggest that SIRT1 deacetylates SIRT6 at K33.

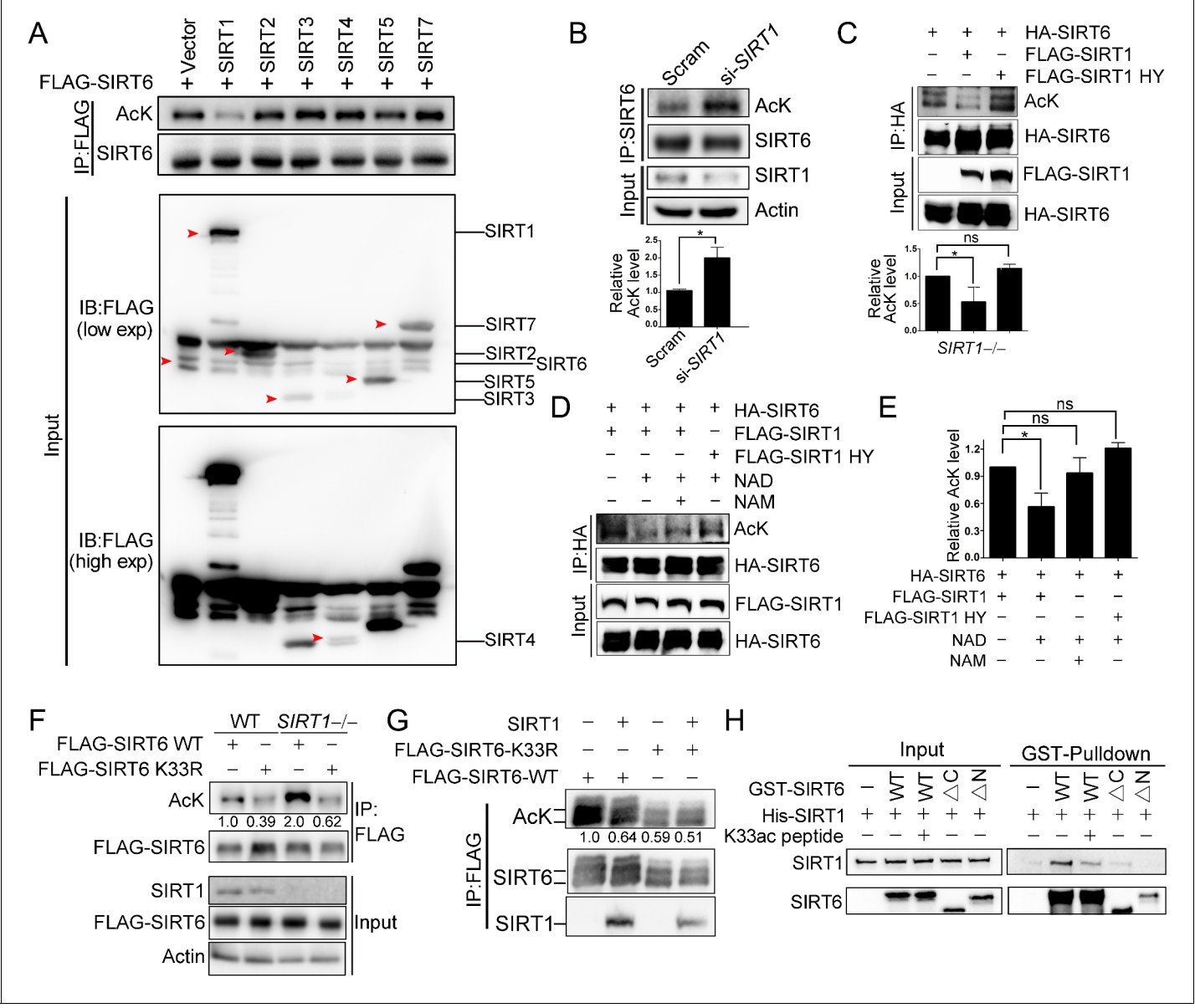

**Figure 4.** SIRT1 deacetylates SIRT6 at K33. (**A**) The acetylation level of FLAG-SIRT6 in HEK293 cells ectopically expressing SIRT1-5 and SIRT7. (**B**) The acetylation level of endogenous SIRT6 in HEK293 cells treated si-*SIRT1* or scramble (Scram) siRNAs. The intensity of acetylated bands was quantified by Image J and normalized to scramble control. The data represent the means ± s.e.m., *p<0.05, n = 3. (**C**) The acetylation level of HA-SIRT6 in *SIRT1*[−/−] cells reconstituted with SIRT1 or the enzyme-inactive SIRT1 H363Y. The intensity of acetylated bands was quantified by Image J and normalized to scramble control. The data represent the means ± s.e.m. *p<0.05, ns: not significant, n = 3. (**D–E**) The acetylation level of HA-SIRT6 in the presence of FLAG-SIRT1, H363Y, NAD$^+$ (500 µM) and/or NAM (2 mM) (**D**). The intensity of acetylated bands was quantified by Image J and normalized to scramble control (**E**).The data represent the means ± s.e.m. *p<0.05, ns: not significant, n = 3. (**F**) The acetylation level of FLAG-SIRT6 and FLAG-SIRT6 K33R in *SIRT1*[−/−] and WT HEK293 cells. (**G**) The acetylation level of FLAG-SIRT6 and K33R in HEK293 cells with or without ectopic SIRT1. (**H**) GST pulldown assay with GST-SIRT6 WT, ΔN, ΔC and His-SIRT1 in the presence or absence of 10 µM K33ac peptide [PEELERK(ac)VWELARL], which represents a 14-aa peptide containing acetylated SIRT6 K33.

The online version of this article includes the following figure supplement(s) for figure 4:

**Figure supplement 1.** Acetylation levels of SIRTs.

## γH2AX ensures SIRT6 retention surrounding DSBs

γH2AX is dispensable for the initial DSB recognition but serves as a platform for recruiting DDR factors (*Celeste et al., 2003*). Because SIRT6 is enriched at DNA breaks, we next asked whether γH2AX is involved in SIRT6 recruitment. We thus performed a co-IP of endogenous SIRT6 in cells treated with or without camptothecin (CPT). Interestingly, H2AX and γH2AX were detected in the anti-SIRT6 precipitates only when the cells were treated with CPT (*Figure 5A,B*). We also performed an in vitro pulldown assay with a biotin-labeled C-terminal γH2AX peptide (biotin-γH2AXp) or H2AX peptide (biotin-H2AXp). Consistently, GST-SIRT6 recognized the γH2AX peptide but not the H2AX peptide (*Figure 5C*). To identify the interacting domain, we purified various GST-SIRT6 truncation mutants. Peptide pulldown assay revealed that the N-terminus truncation was enough to abolish SIRT6 binding to γH2AX peptide, while the C-terminus truncation had a minimal effect (*Figure 5D*). We then investigated whether SIRT1-mediated deacetylation contributes to SIRT6 binding to γH2AX. Here, the K33R mutant efficiently bound to γH2AX to a similar extent as WT SIRT6, but the binding was abolished in the case of K33Q (*Figure 5E*).

To investigate the functional relevance of this SIRT6–γH2AX interaction, we applied laser-induced DNA damage in MEFs lacking *H2ax* and then tracked the distribution of SIRT6 by immunofluorescence microscopy. GFP-SIRT6 was immediately recruited to DNA lesions in *H2ax$^{+/+}$* and *H2ax$^{-/-}$* MEFs (*Figure 5F,G*), implying that H2AX is dispensable for initial SIRT6 recruitment. Interestingly, GFP-SIRT6 diminished from DNA lesions at 10 min after laser treatment in *H2ax$^{-/-}$* MEFs but persisted in *H2ax$^{+/+}$* cells. H2AX is rapidly phosphorylated at serine 139 in response to DSBs (*Rogakou et al., 1998*). When we re-introduced H2AX WT, S139A and S139D into *H2ax$^{-/-}$* MEFs, SIRT6 retention was restored in WT and S139D-re-expressing cells but not in S139A-re-expressing cells (*Figure 5H,I*). Moreover, we used caffeine, an ATM/ATR kinase inhibitor, to treat cells and did SIRT6 recruitment assay. The data showed that caffeine inhibited the level of γH2AX and subsequent retention of SIRT6 at DSBs after laser-induced DNA damage in MEFs (*Figure 5—figure supplement 1*). Together, these data indicate that SIRT6 recognizes γH2AX surrounding DSBs and that this recognition is enhanced by SIRT1-mediated deacetylation.

## SIRT1 and SIRT6 cooperatively promote DNA repair

The physical interaction between SIRT1 and SIRT6 prompted us to further investigate whether SIRT1 and SIRT6 cooperatively modulate the DDR and DNA repair. To do so, we combined the DR-GFP reporter system with a ChIP-PCR analysis. First, we found that FLAG-SIRT6 recruitment to the DSB vicinity was significantly reduced when *SIRT1* was silenced by siRNA in HEK293 cells (*Figure 6A,B*). Then we monitored the dynamic recruitment of GFP-SIRT6 upon laser-induced DNA damage using a MicroPoint system. GFP-SIRT6 was rapidly recruited to DSBs in WT cells, but this process was largely deferred in *Sirt1$^{-/-}$* MEFs (*Figure 6C,D*), suggesting an indispensable role of SIRT1 in the initial recruitment of SIRT6 to DSBs. By contrast, SIRT1 recruitment to DSBs was relatively unaffected by SIRT6 downregulation, as determined by the DR-GFP reporter system (*Figure 6E,F*) and the MicroPoint system (*Figure 6G,H*).

## SIRT6 rescues DNA repair defects caused by SIRT1 deficiency

In our final set of assays, we wanted to determine the function of SIRT6 deacetylation in DNA repair. We found that the interaction of SIRT1 and SIRT6 was enhanced upon DNA damage (*Figure 7A*). In addition, SIRT6 acetylation was significantly decreased upon CPT treatment, but that the effects of CPT were abolished in the presence of SIRT6 K33R or in the absence of *SIRT1* (*Figure 7B,C*). These data imply that SIRT6 is deacetylated by SIRT1 upon DNA damage. We then examined the effects of the SIRT6 mutants on DNA repair by comet assay, which assesses the DNA repair ability at the single cell level. We overexpressed SIRT6 K33R or K33Q in *SIRT6$^{-/-}$* cells and then examined the DNA repair efficacy. Here, overexpression of SIRT6 significantly enhanced DNA repair efficacy upon CPT treatment, while K33Q or H133Y lost the ability. By contrast, the overexpression of SIRT6 K33R promoted DNA repair to an extent comparable to WT (*Figure 7D,E* and *Figure 7—figure supplement 1A*). An HR assay showed that SIRT6 WT and K33R but neither K33Q nor H133Y enhanced HR capacity (*Figure 7F* and *Figure 7—figure supplement 1B*). We further assessed cell viability of *SIRT6$^{-/-}$* HEK293 cells reconstituted with SIRT6 mutants using an MTS assay. The data showed that the viability of cells transfected with SIRT6 WT or K33R was much higher than those transfected with

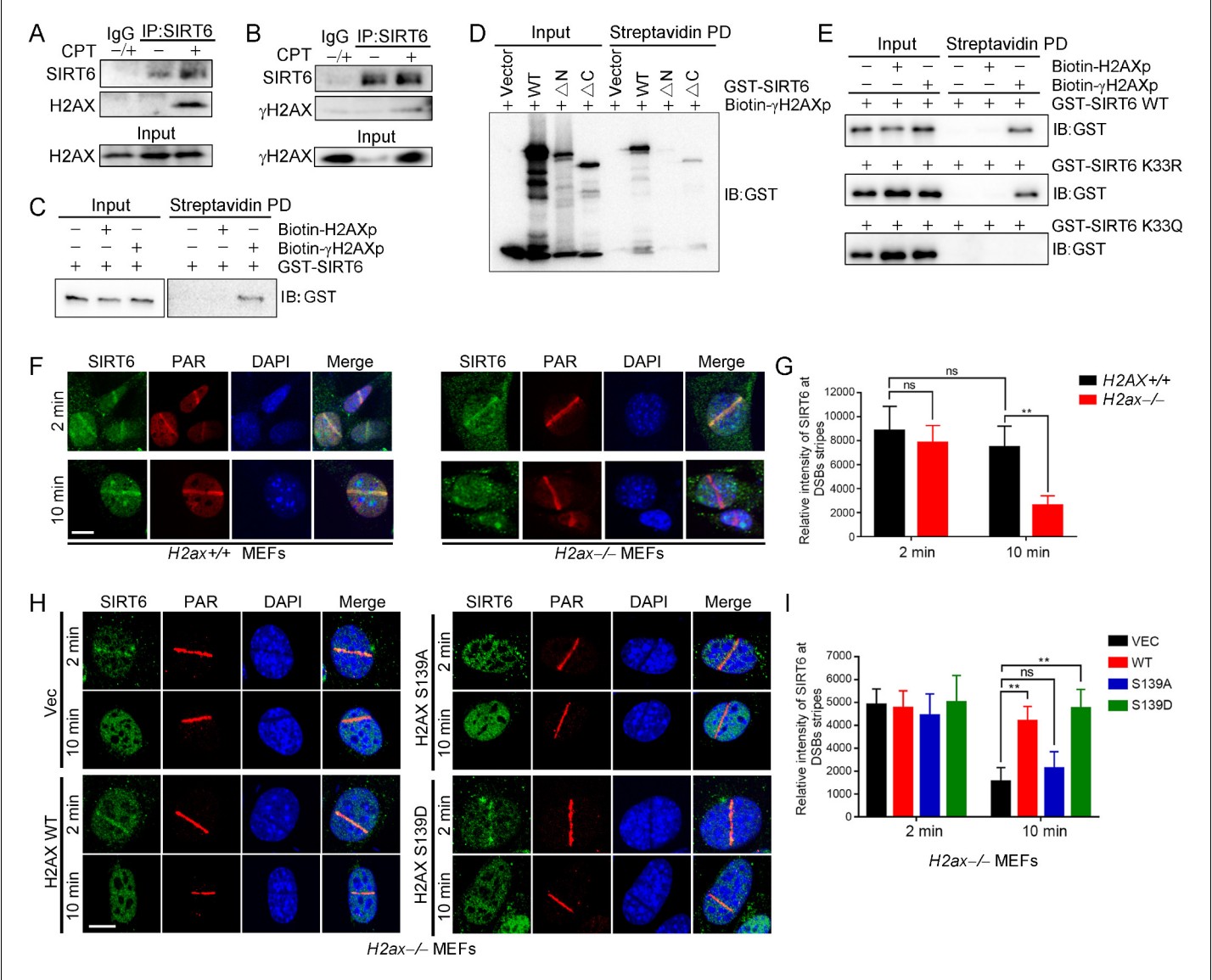

**Figure 5.** γH2AX is required for the chromatin retention of SIRT6. (**A,B**) Representative western blots showing H2AX (**A**) and γ-H2AX (**B**) levels in anti-SIRT6 immunoprecipitates from HEK293 cells treated with or without 1 μM camptothecin (CPT). The IgG control experiment was performed in mixed lysate from cells treated with CPT and cells without CPT. (**C**) Streptavidin pulldown (PD) assay and western blot analysis of the interactions between GST-SIRT6 and biotinylated γH2AX (biotin-γH2AXp) and H2AX peptides (biotin-H2AXp). (**D**) Streptavidin pulldown assay and western blot analysis of the interactions between biotin-γH2AXp, GST-SIRT6 WT and truncated GST-SIRT6 ΔN and ΔC. (**E**) Streptavidin pulldown assay and western blot analysis of the interactions between biotin-γH2AXp and GST-SIRT6 WT, K33R and K33Q. (**F–I**) Laser MicroPoint analysis of SIRT6 recruitment in *H2ax^+/+* and *H2ax^−/−* MEFs (**F–G**), and in *H2ax^−/−* MEFs reconstituted with H2AX WT, S139D mimicking hyper-phosphorylation or S139A mimicking hypo-phosphorylation (**H–I**). PAR immunostaining was used to identify the DNA damage site. Scale bar, 10 μm. The relative fluorescence intensity was calculated by Fiji (Image J) (**G and J**). The data represent the means ± s.e.m., **p<0.01, ns: not significant, n = 10.

The online version of this article includes the following figure supplement(s) for figure 5:

**Figure supplement 1** SIRT6 recruitment in cells treated with caffeine.

empty vector, SIRT6 K33Q or H133Y after CPT or IR treatment (***Figure 7—figure supplement 2***). In HeLa cells, overexpression of SIRT6 K33Q also inhibited the colony-forming capacity upon CPT or IR treatment compared to SIRT6 WT and K33R (***Figure 7—figure supplement 3***). In addition, less γH2AX foci was noticed in HeLa cells transfected ectopic SIRT6 WT or K33R compared to K33Q or H133Y at 8 hr after IR (***Figure 7—figure supplement 4***). These data implicate that SIRT6 deacetylation at K33 is indispensable for DNA repair.

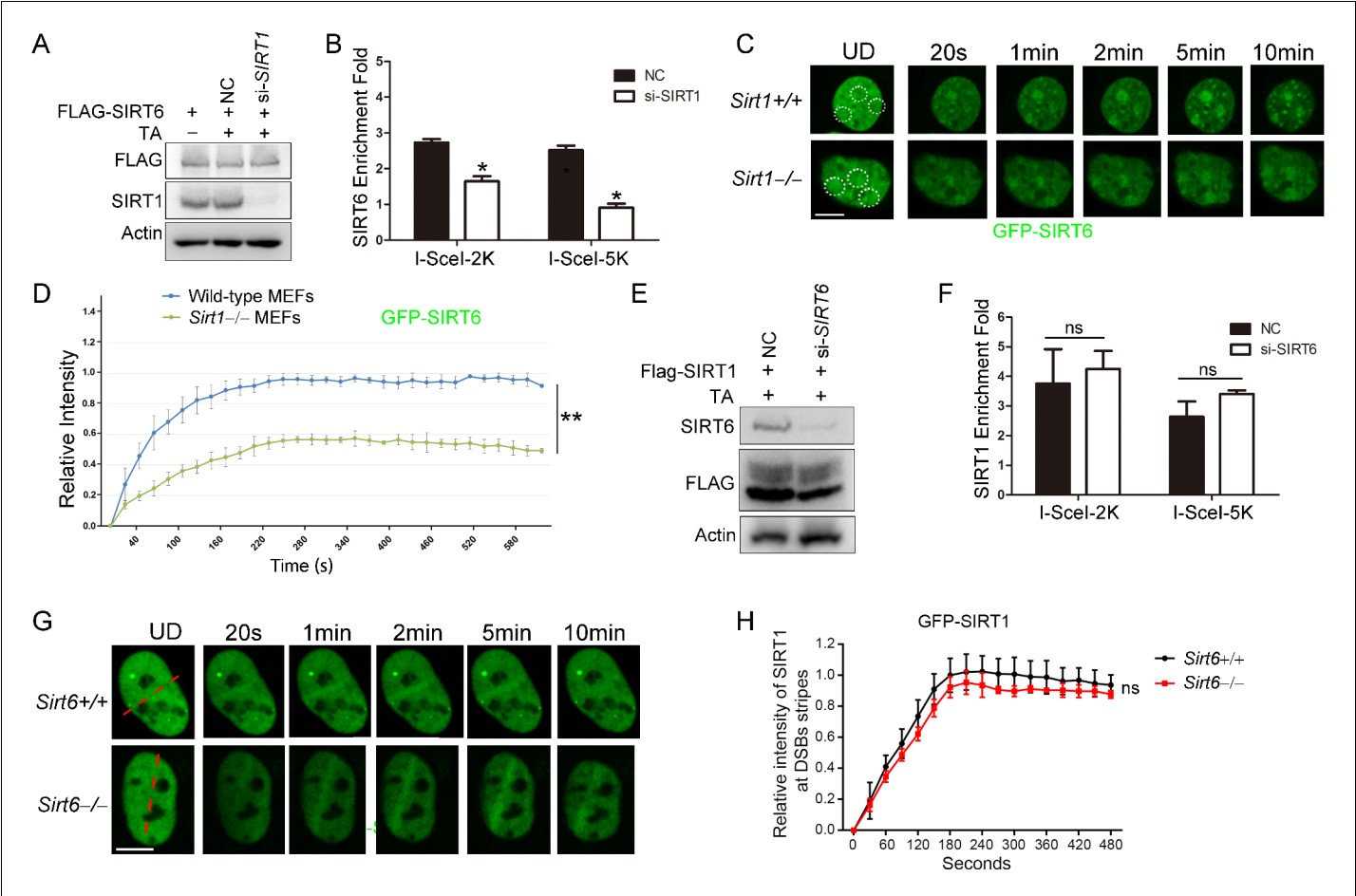

**Figure 6.** SIRT1 facilitates SIRT6 recruitment to chromatin during the DDR. (**A,B**) ChIP-qPCR analysis of the SIRT6 levels in the vicinity of a DSB in cells treated with a *SIRT1* siRNA (si-*SIRT1*) or a scrambled negative control (NC). The western blots show the FLAG-SIRT6 and SIRT1 protein levels. The data represent the means ± s.e.m., *p<0.05, n = 3. (**C,D**) GFP-SIRT6 was introduced into *Sirt1*$^{+/+}$ and *Sirt1*$^{-/-}$ MEFs and the fluorescence signal was captured at various time points after laser-induced DNA damage. Representative images are shown (**C**). The white dashed circles indicate the damage sites. Scale bar, 10 μm. The relative intensity was calculated in Image J (**D**). The data represent the means ± s.e.m., **p<0.01, ns: not significant, n = 30. (**E,F**) ChIP-qPCR analysis of the SIRT1 levels in the vicinity of a DSB in cells treated with *SIRT6* siRNA (*si-SIRT6*) or NC. The western blots show the FLAG-SIRT1 and SIRT6 protein levels. The data represent the means ± s.e.m., ns: not significant, n = 3, determined by Student *t* test. (**G–H**) GFP-SIRT1 was introduced into *Sirt6*$^{+/+}$ and *Sirt6*$^{-/-}$ MEFs and the fluorescence signal was captured after laser-induced damage at various time points. Representative images are shown. The red dashed lines indicate laser-induced damage sites. Scale bar, 10 μm. The data represent the means ± s.e.m., ns: not significant, n = 32.

SIRT1 regulates DNA repair (*Wang et al., 2008*). To elucidate the synergistic effects of SIRTs in DNA repair, we examined whether SIRT6 hyper-acetylation is responsible for the defective DNA repair capacity seen in *SIRT1*$^{-/-}$ HEK293 cells. SIRT6 WT, K33R and SIRT1 overexpression rescued the defective DNA repair imposed by the *SIRT1* deficiency, while SIRT6 K33Q and H133Y had minimal rescue effect (*Figure 7G,H* and *Figure 7—figure supplement 5A*). Notably, both SIRT6 WT and K33R had similar function in rescuing the DNA repair defect in *SIRT1* KO cells, suggesting that over-expressed exogenous SIRT6 WT might not be effectively acetylated. Further, the HR assay showed that SIRT6 WT and K33R, but neither K33Q nor H133Y rescued the HR defect caused by *SIRT1* deficiency (*Figure 7I* and *Figure 7—figure supplement 5B*).

Altogether, these data implicate a synergistic action between SIRT1 and SIRT6 in regulating the DDR and DNA repair. We thus propose a model by which SIRT6 is deacetylated by SIRT1 at K33, thus promoting its polymerization and recognition of DSBs; SIRT6 that is deacetylated at K33 anchors to γH2AX, allowing retention on the chromatin flanking the DSBs and subsequent remodeling via deacetylating H3K9ac and H3K56ac (*Figure 8*).

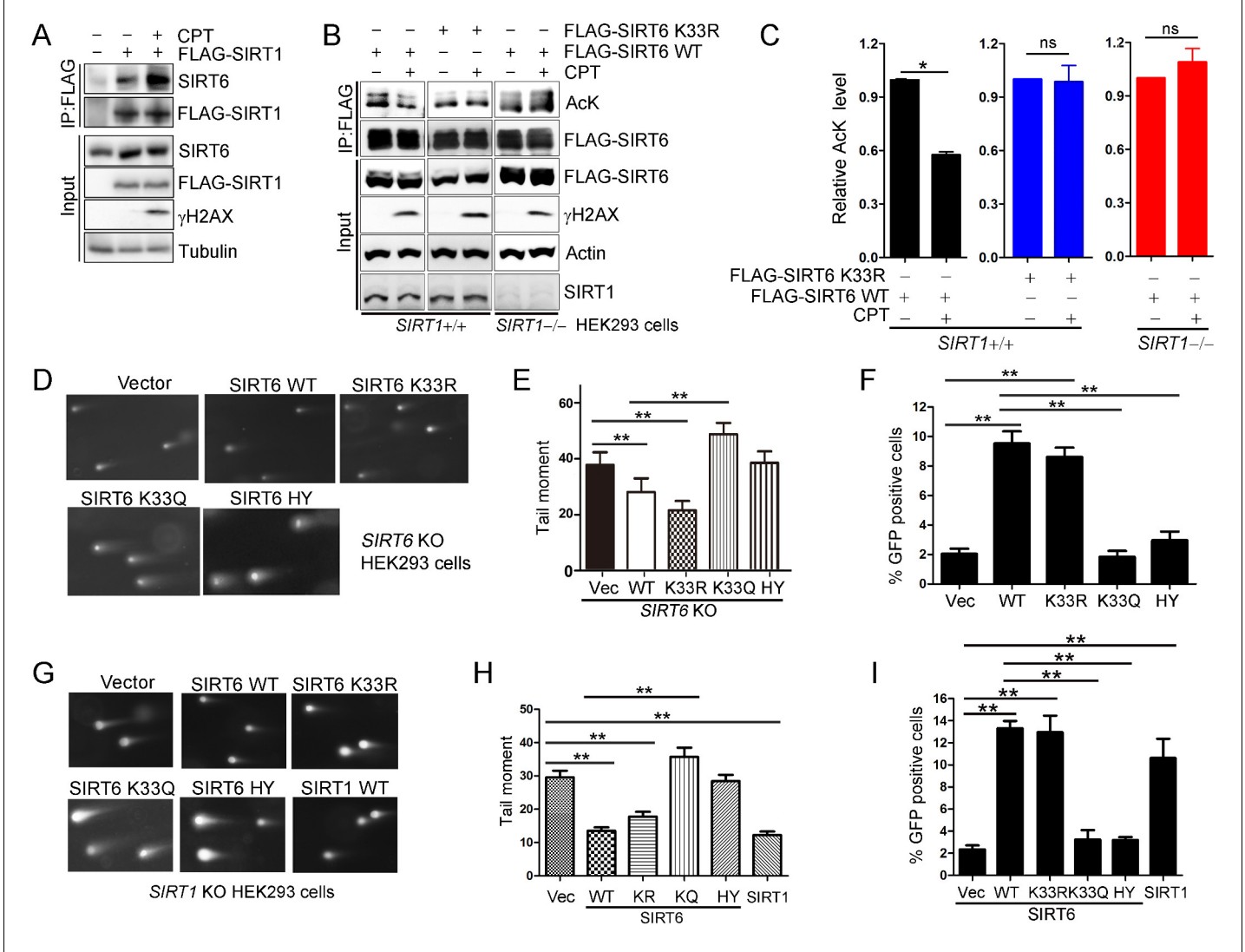

**Figure 7.** SIRT6 rescues DNA repair defects caused by a SIRT1 deficiency. (**A**) Co-IP and western blot analysis of the interaction of FLAG-SIRT1 and SIRT6 in HEK293 cells overexpressing FLAG-SIRT1 and treated with CPT (1 μM) for 1 hr. (**B–C**) The acetylation level of SIRT6 WT and K33R $SIRT1^{+/+}$ and $SIRT1^{-/-}$ HEK293 cells treated or not with CPT (1 μM) for 1 hr. The intensity of acetylated bands was quantified by Image J and normalized to scramble control. The data represent the means ± s.e.m., *p<0.05, ns: not significant, n = 3. (**D–E**) Representative images of comet assay in FLAG-SIRT6, K33R, K33Q and HY reconstituted $SIRT6$ KO cells treated with CPT for 1 hr (**D**). Tail moment were calculated by software Open Comet. The data represent the means ± s.e.m., **p<0.01, n = 50. (**F**) HR assay in U2OS cells ectopically expressing FLAG-SIRT6, K33R, K33Q or HY. The percent GFP-positive cells was calculated. The data represent the means ± s.e.m., **p<0.01, n = 3. (**G–H**) Comet assay in $SIRT1^{-/-}$ HEK293 cells transfected with FLAG-SIRT6, K33R, K33Q, HY and SIRT1 and treated with CPT for 1 hr. Tail moment were calculated by software Open Comet. The data represent the means ± s.e.m., **p<0.01, n = 50. (**I**) HR assay in $SIRT1^{-/-}$ HEK293 cells ectopically expressing FLAG-SIRT6, K33R, K33Q, HY and SIRT1-WY. The percent GFP positive cells was calculated. The data represent the means ± s.e.m., **p<0.01, n = 3.

The online version of this article includes the following figure supplement(s) for figure 7:

**Figure supplement 1.** SIRT6 levels in $SIRT6^{-/-}$ cells.

**Figure supplement 2.** Cell viability assay in $SIRT6^{-/-}$ cells.

**Figure supplement 3.** Colony-forming ability of HeLa cells.

**Figure supplement 4.** γH2AX foci in HeLa cells.

**Figure supplement 5.** SIRTs levels in $SIRT1^{-/-}$ cells.

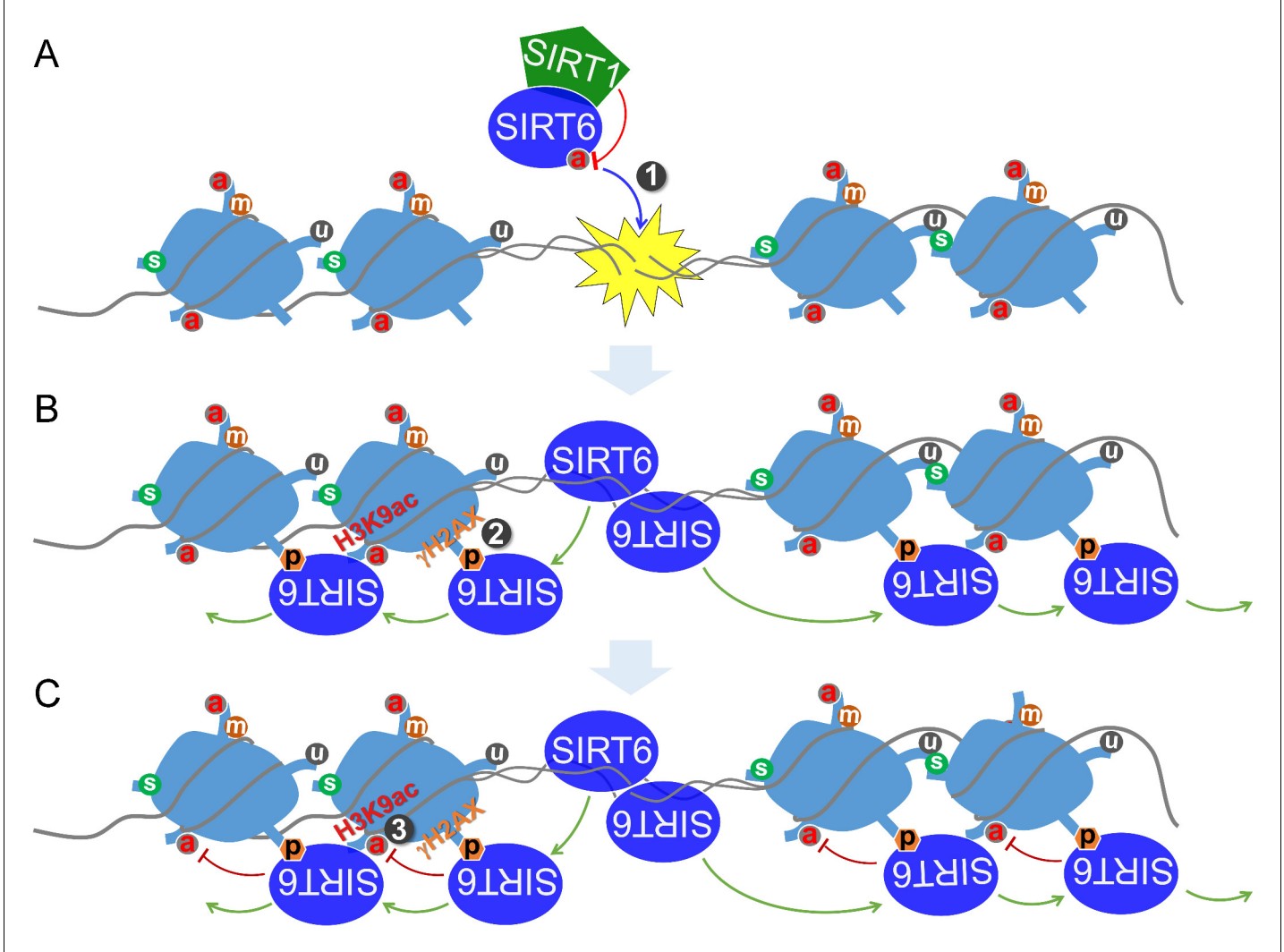

**Figure 8.** A working model. (**A**) SIRT6 is deacetylated by SIRT1 at K33, which promotes SIRT6 polymerization and recognition of DSBs. (**B**) Beyond DSBs, K33-deacetylated SIRT6 anchors to γH2AX and expands on local chromatin flanking DSBs. (**C**) SIRT6 mediates local chromatin remodeling via deacetylating H3K9ac and/or H3K56ac.

## Discussion

The DDR is a highly orchestrated process that is initiated by DNA break-sensing (*Ciccia and Elledge, 2010*). While the MRN complex (*Paull and Lee, 2005*), Ku complex (*Hu et al., 2012*), RPA (*Maréchal and Zou, 2015*) and PARP1 (*Ali et al., 2012*; *Eustermann et al., 2015*) are all known to directly recognize DSBs, sirtuins are among the earliest factors to be recruited to DSBs (*Dobbin et al., 2013*; *Toiber et al., 2013*) and facilitate PARP1 recruitment (*Vazquez et al., 2016*). Consistent with published data (*Pan et al., 2011*), we found that SIRT6 oligomerizes and recognizes DSBs via a DSB-binding pocket generated by the N-termini and C-termini of two adjacent molecules. This finding is consistent with another report showing that both the N-termini and C-termini are essential for the chromatin association of SIRT6 . Using a super-resolution fluorescent particle tracking method, Yang et al. recently found that PARP1 binding to DSBs happens earlier than SIRT6 binding (*Yang et al., 2018*). One possible explanation is that PARP1 is first recruited to DSBs; then, SIRTs are later recruited directly by DSBs and facilitate PARP1 stabilization and expansion in the surrounding region.

The sirtuin family members share similar functions in the DDR and in DNA damage; upon DNA damage, both SIRT1 and SIRT6 are rapidly mobilized to DSBs (*Vazquez et al., 2016*; *Dobbin et al., 2013*; *Toiber et al., 2013*). SIRT1 redistributes on chromatin and deacetylates XPA, NBS1 and Ku70 to promote DNA repair (*Fang et al., 2016*; *Yuan et al., 2007*; *Fan and Luo, 2010*; *Jeong et al., 2007*). Recently, an elegant study demonstrated that PAR recruits SIRT1 and BRG1 to DSB sites and promotes HR efficiency (*Chen et al., 2019*). Other studies reported that SIRT6 mono-ribosylates PARP1 to enhance its activity (*Mao et al., 2011*), and SIRT6 facilitates the subsequent recruitment of SNF2H, H2AX and DNA-PKcs (*Atsumi et al., 2015*; *McCord et al., 2009*; *Van Meter et al., 2016*). Here, we revealed a synergistic action between two nuclear SIRTs in DDR−SIRT1 deacetylates SIRT6 to promote its mobilization to DSBs. A K33R mutant, mimicking the hypo-acetylated SIRT6, can rescue DNA repair defects in *SIRT1* null cells. Both BRG1 and SNF2H are chromatin remodeling ATPases, responsible for open chromatin architecture. It is reasonable to speculate that these early DDR responding factors like PARP1, SIRT1, SIRT6, SNF2H and BRG1 are quickly and sequentially stimulated by DSBs, wherein they constitute a super complex to potentiate DDR and DNA repair; posttranslational modifications like deacetylation and mono-ADP ribosylation empower the complex to recruit other repair factors more efficiently. Interestingly, SIRT6 phosphorylation at S10 by JNK promotes subsequent recruitment itself and PARP1 upon oxidative stress, also supporting an essential role of the SIRT6 N terminus for DSB-recruitment (*Van Meter et al., 2016*). Consistent with the cooperative action between SIRT1 and SIRT6, independent studies have revealed an interaction between SIRT1 and SIRT7, showing that SIRT1 recruits SIRT7 to promote cancer cell metastasis (*Malik et al., 2015*), and that SIRT1 and SIRT7 antagonistically regulate adipogenesis (*Fang et al., 2017*).

The acetylation levels of H3K9 and H3K56 decrease upon detecting DSBs and then return to basal levels (*Tjeertes et al., 2009*). SIRT1 and SIRT6 are H3K9ac and H3K56ac deacetylases; both are recruited to DSBs, indicating that SIRT1 and/or SIRT6 might contribute to reducing H3K9ac and H3K56ac levels. Although mechanistically unclear, H3K9ac and H3K56ac levels negatively correlate with γH2AX levels (*Tjeertes et al., 2009*). In this study, we found that while γH2AX is not required for initial SIRT6 recruitment, it is indispensable for retaining SIRT6 on the local chromatin surrounding DSBs. This finding is consistent with reports that γH2AX is dispensable for initial reorganization of DNA breaks but rather serves as a platform to stabilize DNA repair factors, such as NBS1, 53BP1 and BRCA1. SIRT6 deacetylates H3K9ac and H3K56ac surrounding DSBs, in this way bridging γH2AX to chromatin remodeling. While the in vivo data demonstrated that SIRT6 K33Q deacetylation activity toward histone H3 was compromised, the in vitro deacetylation assay using a synthesized acetyl H3 peptide showed negligible effect. It is speculated that the initial DSB recognition and chromatin retention might potentiate the deacetylase activity of SIRT6 toward local histones, for example H3K9ac and H3K56ac; the impaired DSB recognition and chromatin retention might compromise the deacetylase activity of SIRT6 K33Q on local histone proteins. Putting together the findings provide a scenario as to how γH2AX and histone modifiers coordinate to amplify the DDR.

SIRT6 and SNF2H cooperate to stabilize γH2AX foci (*Atsumi et al., 2015*). Here we found that γH2AX in-fact anchors SIRT6 to DSBs, providing a positive feedback regulatory loop between SIRT6 and γH2AX. This finding is consistent with reports showing a distinct reduction of γH2AX and an improper DDR in *Sirt6*$^{-/-}$ and *Sirt1*$^{-/-}$ cells. Recent work also suggests that an electrostatic force between a negatively charged phosphate group and a positively charged lysine groups is a novel form of protein–protein interaction (*Wang et al., 2016*). We thus consider it plausible to speculate that (de)acetylation might act as a switch to modulate such an interaction between SIRT6 and γH2AX.

Known as longevity-associated genes, SIRT6 and SIRT1 are redundant in DNA repair but not replaceable. In this study, we have identified that SIRT6 directly binds to DNA breaks and have elucidated a physical and functional interaction between SIRT6 and SIRT1. SIRT6 rescues DNA repair defects imposed by SIRT1 deficiency. Overall, these data highlight a synergistic action of nuclear SIRTs in the spatiotemporal regulation of the DDR and DNA repair.

# Materials and methods

**Key resources table**

| Reagent type (species) or resource | Designation | Source or reference | Identifiers | Additional information |
|---|---|---|---|---|
| Gene (*Homo sapiens*) | SIRT6 | National Center for Biotechnology Information | Gene ID: 51548 | |
| Gene (*Homo sapiens*) | SIRT1 | National Center for Biotechnology Information | Gene ID: 23411 | |
| Gene (*Mus musculus*) | H2ax | National Center for Biotechnology Information | Gene ID: 15270 | |
| Gene (*Homo sapiens*) | H2AX | National Center for Biotechnology Information | Gene ID: 3014 | |
| Cell line (*Homo sapiens*) | HEK293 | ATCC | ATCC CRL-1573 | |
| Cell line (*Homo sapiens*) | HeLa | ATCC | ATCC CRM-CCL-2 | |
| Cell line (*Mus musculus*) | MEF | Dr Linyu Lu (Zhejiang University, China) | | |
| Cell line (*Mus musculus*) | H2ax-/- MEF | Dr Linyu Lu (Zhejiang University, China) | | |
| Antibody | SIRT6 (rabbit, polyclonal) | Abcam (Cambridge, UK) | Cat# ab62738, RRID:AB_956299 | Applications: WB; Dilution: 1:1000; Immunofluorescence; Dilution:1:100 |
| Antibody | SIRT1 (mouse, monoclonal) | Cell Signaling Technology | Cat# 8469, RRID:AB_10999470 | Applications: WB; Dilution:1:1000; Immunofluorescence; Dilution:1:100 |
| Antibody | FLAG (mouse, monoclonal) | Sigma-Aldrich | Cat# F1804; RRID:AB_262044 | Applications: WB; Dilution: 1:1000; Chromatin immuno precipitation |
| Antibody | HA (mouse, monoclonal) | Sigma-Aldrich | Cat# H3663; RRID:AB_262051 | Applications: WB; Dilution: 1:1000 |
| Antibody | GST (mouse, monoclonal) | Cell Signaling Technology | Cat# 2624, RRID:AB_2189875 | Applications: WB; Dilution: 1:1000 |
| Antibody | γH2AX (rabbit, monoclonal) | Abcam (Cambridge, UK) | Cat# ab81299; RRID:AB_1640564 | Applications: WB; Dilution: 1:1000 |
| Antibody | H3K9ac (rabbit, polyclonal) | EMD Millipore | Cat# 07–352; RRID:AB_310544 | Applications: WB; Dilution: 1:1000 |
| Antibody | H3K56ac (Rabbit, Polyclonal) | EMD Millipore | Cat# 07–677, RRID:AB_390167 | Applications: WB; Dilution: 1:1000 |
| Antibody | acetyl Lysine (Rabbit, Polyclonal) | Abcam (Cambridge, UK) | Cat# ab80178, RRID:AB_1640674 | Applications: WB; Dilution: 1:1000 |
| Transfected construct (*Homo sapiens*) | pDR-GFP | Addgene (Cambridge, MA) | RRID:Addgene_26475 | |
| Commercial assay or kit | CycLex SIRT6 Deacetylase Fluorometric Assay Kit | MBL life science | CY-1156V2 | |
| Chemical compound, drug | Ex527 | Sigma-Aldrich | E7034 | |
| Chemical compound, drug | Trichostatin A | Sigma-Aldrich | T1952 | |
| Chemical compound, drug | Nicotinamide | Sigma-Aldrich | N3376 | |

*Continued on next page*

*Continued*

| Reagent type (species) or resource | Designation | Source or reference | Identifiers | Additional information |
|---|---|---|---|---|
| Chemical compound, drug | Camptothecin | Sigma-Aldrich | C9911 | |
| Software, algorithm | GraphPad Prism | GraphPad | RRID:SCR_002798 | |

## Cell lines

HEK293 (CRL-1573) cells and HeLa (CCL-2) cells were ordered from ATCC. *H2ax* WT and KO mouse embryonic fibroblasts (MEFs) were provided as a kind gift from Dr Linyu Lu (Zhejiang University, China). The cell lines were authenticated by short tandem repeat (STR) profile analysis and genotyping and were mycoplasma free. Cells were routinely cultured in Gibco High Glucose DMEM (Life Technologies, USA) with 10% fetal bovine serum (FBS), 100 U/ml penicillin and streptomycin (P/S) at 37°C in 5% $CO_2$ and atmospheric oxygen conditions.

## Oligos and plasmids

The following oligos (Genewiz) were used for RNA interference:

siSIRT6, 5'-AAGAAUGUGCCAAGUGUAAGA-3';
siSIRT1, 5'-ACUUUGCUGUAACCCUGUA-3'.

The following primers were used for ChIP qPCR:

I-SceI- 2 k-F, 5'-GCCCATATATGGAGTTCCGC-3';
I-SceI-2k-R, 5'-GGGCCATTTACCGTCATTG-3';
I-SceI-5k-F, 5'-GTTGCCGGGAAGCTAGAGTAAGTA-3';
I-SceI-5k-R, 5'-TTGGGAACCGGAGCTGAATGAA-3'.

The following gRNA sequences were used for CRISPR/Cas9 gene editing:

Hu Sirt6: gRNA-F, 5'-CACCGGCTGTCGCCGTACGCGGACA-3';
gRNA-R, 5'-AAACTGTCCGCGTACGGCGACAGCC-3'.
Hu Sirt1: gRNA-F, 5'-CACCGATAGCAAGCGGTTCATCAGC-3'

Human SIRT6 was cloned into pCDNA3.1 with a FLAG tag (Invitrogen, USA); a 3 × FLAG-SIRT1 and DR-GFP plasmids were obtained from Addgene. SIRT6ΔC and ΔN were amplified with specific primers and cloned into pKH3HA (Addgene) and pGex vectors (GE Healthcare Life Sciences). The SIRT6 KR, KQ and HY mutants were obtained by converting SIRT6 lysine 33 to arginine (KR), or to glutamine (KQ) and SIRT6 133 histidine to tyrosine (HY) via site-directed mutagenesis, as described below.

## Site-directed mutagenesis

The primers used for mutagenesis were designed using the online Quick Change Primer Design Program provided by Agilent Technologies. The mutagenesis was performed using Pfu DNA polymerase (Agilent) and 300 ng plasmid template, according to the manufacturer's instructions. The PCR product was digested with *Dpn*I endonuclease for 1 hr at 37°C, before transformation and sequencing.

The following primers were used to generate the SIRT6 KR, KQ and HY mutants:

KR forward: 5'-ggagctggagcggagggtgtgggaact-3'
KR reverse: 5'-agttcccacaccctccgctccagctcc-3'
KQ forward: 5'-ggagctggagcggcaggtgtgggaact-3'
KQ reverse: 5'-agttcccacacctgccgctccagctcc-3'
HY forward: 5'-acaaactggcagagctctacgggaacatgtttgtg-3'
HY reverse: 5'-cacaaacatgttcccgtagagctctgccagtttgt-3'

## Immunoprecipitation

HEK293T cells were transfected with the indicated plasmids using Lipofetamine3000 (Invitrogen, USA), according to the manufacturer's instructions. The cells were lysed 48 hr post-transfection in lysis buffer [50 mM Tris-HCl, pH 7.4, 200 mM NaCl, 0.2% NP40, 10% glycerol, 1 mM NaF, 1 mM Sodium butyrate, 10 mM Nicotinamide and a Complete protease inhibitor cocktail (Roche)]. The cell extracts were incubated with anti-FLAG M2 monoclonal antibody-conjugated agarose beads (Sigma) at 4℃ overnight with rotation. The immunoprecipitates were boiled IN 2 × laemmli buffer and then analyzed by western blotting.

## Chromatin immunoprecipitation (ChIP)

I-*Sce*I-GR assays were performed as previously described (*Soutoglou et al., 2007*). HeLa cells stably transfected with DR-GFP were transiently transfected with RFP-I-*Sce*I-GR together with FLAG-SIRT6, KR, KQ or HY. The cells were treated with $10^{-7}$ M triamcinolone acetonide (TA, Sangon, Shanghai) for 20 min, 48 hr after transfection, and then fixed with 1% paraformaldehyde at 37℃ for 10 min to crosslink the chromatin. The reaction was stopped upon the addition of 0.125 M glycine. The chromatin was sonicated to 200 bps ∼ 600 bps and incubated with the indicated antibodies. After de-cross linking, the ChIP-associated DNA was isolated and analyzed by quantitative real-time PCR (qRT-PCR).

## Comet assay

A comet assay was performed as previously described (*Olive and Banáth, 2006*). Briefly, after CPT treatment, the cells were digested into a single cell suspension, mixed with 1% agarose at the density of $1 \times 10^5$, coated on the slide and then incubated in lysis buffer (2% sarkosyl, 0.5M $Na_2$EDTA, 0.5 mg/ml proteinase K) overnight at 37℃. The slides were incubated with N2 buffer (90 mM Tris, 90 mM boric acid and 2 mM $Na_2$EDTA) and subjected to electrophoresis for 25 min at 0.6 V/cm. The slides were then incubated in staining solution containing 2.5 μg/ml propidium iodide for 30 min at room temperature. Images were captured under a fluorescent microscope.

## Cell fractionation

Cells were scraped and washed with cold PBS. The cell pellet was resuspended in nuclei lysis buffer (10 mM HEPES, 10 mM KCl, 1.5 mM $MgCl_2$, 0.34M sucrose, 10% glycerol, 1 mM DTT, 0.1% TrionX-100.) for 10 min on ice and then centrifuged at 1300 *g* for 10 min. The pellet was resuspended in lysis buffer (3 mM EDTA, 0.2 mM EGTA, 1 mM DTT) for 10 min on ice and centrifuged at 1700 *g* for 10 min. The pellet was saved as the chromatin fraction.

## MicroPoint laser irradiation and microscopy

U2OS cells or MEFs were seeded on a dish with a thin glass bottom (NEST), then locally irradiated with a 365 nm pulsed UV laser (16 Hz pulse, 56% laser output), generated by the MicroPoint Laser Illumination and Ablation System (Andor; power supply TPES24-T120MM, Laser NL100, 24V 50W), which is coupled to the fluorescence path of the Nikon A1 confocal imaging system (TuCam). Fluorescent protein recruitment and retention were continuously monitored by time-lapse imaging every 20 s for 10 min. The fluorescence intensity was quantified at each time-point using Fiji (Image J) software.

## CRISPR/Cas9-mediated gene editing

CRISPR/Cas9-mediated gene editing was conducted as described (*Ran et al., 2013*). Briefly, a pX459 vector (Addgene#48139) was digested with *Bbs*I and ligated with annealed oligonucleotides. The constructs containing the target gRNAs were transfected into HEK293T cells with Lipofetamine3000 (Invitrogen). The cells were selected for 5 days with puromycin 24 hr after transfection. Single clones were picked for sequencing.

## Peptide pulldown assay

The C termini of H2AX (BGKKATQASQEY) and γH2AX (BGKKATQApSQEY) were synthesized and conjugated with biotin (GL Biochem, Shanghai). For one reaction, 1 μg biotinylated peptides was incubated with 1 μg GST-SIRT6 in binding buffer (50 mM Tirs-HCl, 200 mM NaCl, 0.05% NP40)

overnight at 4°C. Streptavidin Sepharose beads (GE) was then used to pulldown the peptide and protein complexes for 1 hr at 4°C, and the samples were analyzed by western blotting.

## Immunofluorescence staining

The cells were washed with PBS and fixed with 4% formaldehyde for 20 min, followed by permeabilization with cold methanol (−20°C) for 5 min and blocking with 5% BSA for 30 min. Then, the cells were incubated with primary antibodies (SIRT1, 1:200 dilution in 1% BSA; γH2AX, 1:500 dilution in 1% BSA; SIRT6, 1:200 dilution in 1% BSA) for 1 hr and secondary antibodies (donkey anti-rabbit IgG Alexa Fluor 594 and donkey anti-mouse IgG FITC from Invitrogen, 1:500 dilution in1% BSA) for 1 hr at room temperature in the dark. The cells were then co-stained with DAPI (Invitrogen) and observed under a fluorescent microscope.

## HR assay

U2OS cells stably transfected with DR-GFP were transfected with HA-I-SceI together with FLAG-SIRT6 WT, K33R, K33Q or H133Y. After transfection for 48 hr, the cells were harvested and the GFP-positive cell ratio per $10^4$ cells was analyzed by flow cytometry (BD Biosciences). The relative HR efficiency was normalized to the vector control.

## Assessment of cell viability by MTS assay

Cell proliferation rate was examined using a CellTiter 96 AQueous Non-Radioactive Cell Proliferation assay (MTS) (Promega, USA). Approximately $2 \times 10^3$ cells/well were seeded in a 96-well plate and allowed to grow overnight. Cells were treated with CPT (0, 1 μM) for 1 hr or IR (2 Gy, 4 Gy, 6 Gy) and allowed to recover for 48 hr. MTS reagent (20 μL per well) was added, followed by incubation in the darkness at 37°C for 3 hr. The absorbance were measured at 490 nm using Bradford Reagent (Bio-Rad Laboratories). Cell viability was calculated as the ratio of absorbance of treated cells to control.

## Colony formation assay

HeLa cells were seeded into six-well plates 24 hr after transfection in defined numbers. Then, 24 hr after re-plating, the cells were exposed to increasing amounts of ionizing radiation delivered by an X-Rad 320 irradiator (Precision X-Ray Inc N. Branford, CT, USA). Fresh media was added after 7 days. Colonies containing at least 50 cells (10–14 days) were fixed with 20% methanol and stained with crystal violet and analyzed.

## DNA pulldown assay

The DNA binding assay was performed as previously described (*Falck et al., 2005*). Briefly, a biotin-conjugated DNA duplex 220 bp in size was generated by PCR amplification using biotin-labeled primers and a I-SceI plasmid as a template.

For the DNA pulldown assay, 10 pmol biotinylated DNA duplex was incubated with 0.5 μg of the indicated recombinant proteins in 300 μl binding buffer (10 mM Tris-Cl pH7.5, 100 mM NaCl, 0.01% NP40 and 10% glycerol) overnight at 4°C. Streptavidin Sepharose beads (GE) were added the next day, and incubated for another 1 hr with the samples. The beads were then collected and washed with binding buffer three times. The beads were subsequently boiled in $2 \times$ laemmli buffer and analyzed by western blotting.

For linear and circular DNA competition assays, the ratios of the non-biotin labeled linear/circular DNA to the biotin DNA duplex were 5:1 or 10:1. Linear DNA was generated by PCR amplification using non-biotin-labeled primers, and circular DNA was obtained by cloning a PCR product into the pCDNA 3.1 plasmid (Invitrogen, USA).

The following sequences were used for PCR:

Forward, 5'-TACGGCAAGCTGACCCTGAA-3'
Reverse, 5'-CGTCCTCCTTGAAGTCGATG-bio-3'

## FP assay

SIRT1, SIRT6 and SIRT7 recombinant proteins were purified in vitro, and incubated with a FAM-conjugated DNA duplex (20 nM) for 30 min on ice at the indicated concentration. The FP value of each sample was measured on 96 plates using a Multimode Plate Reader VictorTM X5 (PerkinElmer, USA) with an excitation wavelength of 480 nm and an emission wavelength of 535 nm. Curve fitting was performed in GraphPad prism.

## Statistical analysis

Statistical analyses were conducted using two-tailed Student's *t*-test between two groups. All data are presented as mean ± s.e.m. as indicated, and a p value < 0.05 was considered statistically significant. Independent experiments were performed in triplicates.

## Acknowledgements

We thank Dr Linyu Lu (Zhejiang University, China) for providing *H2ax*$^{-/-}$ MEFs. This project was supported by research grants from the National Key R and D Program of China (2017YFA0503900), the National Natural Science Foundation of China (91849208, 81972602, 81702909, 81871114, 81601215, 91949124, 31530016, 31761133012), the National Natural Science Foundation of Guangdong Province (2015A030308007, 2017B030301016), Shenzhen Science and Technology Innovation Commission (ZDSYS20190902093401689, KQJSCX20180328093403969, JCYJ20180507182044945, JCYJ20180507182213033, JCYJ20170412113009742), the Youth Foundation of Tianjin Medical University Cancer Institute and Hospital (NO. B1714), and Tianjin Municipal Science Foundation for Youths (NO. 18JCQNJC79800). The authors would like to thank Dr. Jessica Tamanini (ETediting, Shenzhen University) for editing the manuscript prior to submission

## Additional information

### Funding

| Funder | Grant reference number | Author |
|---|---|---|
| National Key R&D Program of China | 2017YFA0503900 | Wei-Guo Zhu<br>Baohua Liu |
| National Natural Science Foundation of China | 91849208 | Baohua Liu |
| National Natural Science Foundation of China | 91949124 | Minxian Qian |
| National Natural Science Foundation of China | 81972602 | Xiaolong Tang |
| National Natural Science Foundation of China | 81702909 | Xiaolong Tang |
| National Natural Science Foundation of China | 81871114 | Minxian Qian |
| National Natural Science Foundation of China | 81601215 | Zuojun Liu |
| Natural Science Foundation of Guangdong Province | 2015A030308007 | Baohua Liu |
| Natural Science Foundation of Guangdong Province | 2017B030301016 | Minxian Qian<br>Wei-Guo Zhu<br>Xingzhi Xu<br>Baohua Liu |
| Shenzhen Science and Technology Innovation Commission | ZDSYS20190902093401689 | Baohua Liu |
| Shenzhen Science and Technology Innovation Commission | JCYJ20180507182044945 | Baohua Liu |

| Shenzhen Science and Technology Innovation Commission | KQJSCX20180328093403969 | Baohua Liu |
|---|---|---|
| Tianjin MunicipalScience Foundation for Youths | 18JCQNJC79800 | Fanbiao Meng |
| Tianjin Medical University | B1714 | Fanbiao Meng |
| National Natural Science Foundation of China | 31530016 | Xingzhi Xu |
| National Natural Science Foundation of China | 31761133012 | Xingzhi Xu |
| Shenzhen Science and Technology Innovation Commission | JCYJ20180507182213033 | Xingzhi Xu |
| Shenzhen Science and Technology Innovation Commission | JCYJ20170412113009742 | Xingzhi Xu |

The funders had no role in study design, data collection and interpretation, or the decision to submit the work for publication.

## Author contributions

Fanbiao Meng, Data curation, Formal analysis, Validation, Investigation, Methodology, Writing - original draft; Minxian Qian, Data curation, Formal analysis, Validation, Investigation, Methodology; Bin Peng, Data curation, Investigation, Methodology; Linyuan Peng, Xiaohui Wang, Zuojun Liu, Xiaolong Tang, Investigation, Methodology; Kang Zheng, Software, Visualization, Methodology; Shuju Zhang, Shimin Sun, Methodology; Xinyue Cao, Methodology, Project administration; Qiuxiang Pang, Bosheng Zhao, Resources; Wenbin Ma, Zhou Songyang, Bo Xu, Resources, Methodology; Wei-Guo Zhu, Resources, Funding acquisition; Xingzhi Xu, Resources, Software, Funding acquisition, Methodology; Baohua Liu, Conceptualization, Supervision, Funding acquisition, Writing - review and editing

## Author ORCIDs

Fanbiao Meng https://orcid.org/0000-0002-1227-9390
Minxian Qian http://orcid.org/0000-0002-1763-2325
Kang Zheng https://orcid.org/0000-0002-6347-4241
Xiaolong Tang http://orcid.org/0000-0002-4744-5846
Wei-Guo Zhu http://orcid.org/0000-0001-8385-6581
Baohua Liu https://orcid.org/0000-0002-1599-8059

## Decision letter and Author response

Decision letter https://doi.org/10.7554/eLife.55828.sa1
Author response https://doi.org/10.7554/eLife.55828.sa2

# Additional files

## Supplementary files

- Supplementary file 1. Acetylated K residues of SIRT6 identified by LC-MS/MS.
- Transparent reporting form

## Data availability

All data generated or analysed during this study are included in the manuscript and supporting files.

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
