## [Decision Letter]

**Acceptance summary:**

The work presented goes to a great deal of effort to succinctly demonstrate a synergistic interaction between two predominant proteins involved with DNA repair. These two sirtuins are of great interest in several fields, including cancer, aging, and genome stability. Previous research has largely focused on one or the other, often times noting overlaps in function and pathway, but rarely has there been much insight into how these two proteins interact. This manuscript provides some much needed insight into the relationship between sirtuins and how they manage the critical process of DNA repair.

**Decision letter after peer review:**

Thank you for submitting your article "Synergy between SIRT1 and SIRT6 helps recognize DNA breaks and potentiate the DNA damage response and repair" for consideration by *eLife*. Your article has been reviewed by three peer reviewers, including Matthew Simon as the Reviewing Editor and Reviewer #1, and the evaluation has been overseen by Jessica Tyler as the Senior Editor.

The reviewers have discussed the reviews with one another and the Reviewing Editor has drafted this decision to help you prepare a revised submission. Overall, the work was well received and the reviewers are quite positive that the work will be impactful to the field once the requested revisions are addressed. Please aim to submit the revised version within two months. If you require a longer length of time due to extenuating circumstances with the virus pandemic, please don't hesitate to ask.

Summary:

In Meng, et al., the authors provide a compelling body of evidence to support the observations that SIRT1 deacetylates SIRT6 in order to facilitate SIRT6 polymerization and subsequent recognition of DSB. This, in turn, helps to recruit repair proteins and stabilize SIRT6 at DNA damage sites. The authors demonstrate that SIRT6 is able to directly recognize DNA DSBs which supports a recent study published in *eLife* (Toiber et al., 2020). Interestingly, the authors describe a novel mechanism whereby SIRT6 needs to be deacetylated in order to recognize DNA DSBs and that this deacetylation is performed by SIRT1. Through the use of SIRT6 acetyl mutants, the authors map the acetylation site to K33 and demonstrate through several robust experiments the interaction between this site and SIRT1. The authors then describe experiments that suggest the DNA damage-induced histone mark gH2AX is dispensable for the initial recruitment of deacetylated SIRT6, but required for its retention at DSB sites. The authors finalize the manuscript by suggesting that expression of a hypoacetylated SIRT6 (K33R) is able to overcome DNA repair defects caused by SIRT1 deficiency.

Essential revisions:

1) In Figure 2C, the authors suggest that SIRT6 K33R does indeed have increased DSB affinity when compared to the K33Q mutant, and that this is supportive of the data presented in Figure 2B. Whilst it is true that K33R binds DNA more efficiently in this assay then K33Q, it is not possible to interpret the significance of the hypoacetylation for this binding without WT SIRT6. The authors need to repeat this experiment including recombinant WT SIRT6 as a control. One would expect that wild type SIRT6 DNA binding affinity should be somewhere in between the two mutants.

2) In the text the authors state that the enzymatic activity of the SIRT6 K33Q is unaffected. However, based on the IF data in Figure 2 as well as the histone acetyl western blots in Figure 2—figure supplement 3, this does not seem to be the case. Can the authors please rationalize the differences seen in the IF and immunoblot experiments compared to their enzymatic activity assay described in Figure 2—figure supplement 3C.

3) For Figure 3G, the authors need to address the lack of pull down of the c-terminal peptide. The authors concluded that N terminus of SIRT6 is crucial for SIRT1-SIRT6 interaction. However, in Figure 3G, both SIRT6ΔN and ΔC fail to interact with SIRT1. Although the authors claimed that the result arises from the nuclear location signal is located in C terminus, a more direct evidence such as in vitro immunoprecipitation of SIRT6ΔC with SIRT1 is recommended.

4) The biochemistry experiments performed in Figure 3 are nicely done. Do the authors see an increase in the interaction between SIRT6 and SIRT1 upon DNA damage (e.g. Ionizing Radiation)? This would be a strong experiment to include.

5) Figure 4 needs some clarification with labeling and constructs being used. Labeling in 4A is unclear. In Figures 4C and 4D the authors ectopically express Flag-tagged SIRT1 and SIRT6 constructs. They then perform a Flag IP to isolate SIRT6 to determine the acetylation levels of SIRT6 in the presence of wild type SIRT1, enzymatically inactive SIRT1 or in the presence of inhibitors (Figure 4D). This makes interpretation of this data difficult, as it's not clear how the authors know which sirtuin is being immunoprecipitating when both are flag-tagged and the immunoprecipitation is being performed using a flag antibody. These experiments need to be repeated with a different tag on either SIRT1 or SIRT6 if this is the case. In Figure 4E, is the flag immunoblot the input or the IP? Going by the labelling, it appears as if this is the input. If this is the case, then there is no FLAG IP immunoblot shown and interpretation of the AcK immunoblot is inhibited without this data. Please provide the Flag IP immunoblot for this experiment.

6) Explain the elevated γH2AX level in the absence of CPT in Input of Figure 5B.

7) Several gel images need to be repeated for clarity, as the conclusions drawn from them are critical to the thesis, but the ability to discern the accuracy of the gel images is weak. In particular, Figure 2G, 4D, 5E and 7A. Figure 7A, in particular, is very unclear (SIRT1 ^-/-^ bands look to be different sizes compared to the other lanes).

8) Many gel images and subsequent assays lack quantification. All images and laser experiments should include quantification to more accurately present the observations, as well as to provide the statistical rigor used. The graphics of Western blot need to have a uniform format (some with borders, some without). In regards to statistics, the authors need to clarify the n-values used for the vast majority of their experiments.

9) Using the ISce1 system, the authors demonstrate that K33R is as effective as recognizing DNA DSBS as wild type SIRT6, and far superior to K33Q and HY mutants. One would therefore expect that the DNA repair kinetics of K33R is far superior to K33Q and HY mutants, and at least comparable to SIRT6 wild type. Can the authors show this by performing immunofluorescence staining/foci analysis for key DNA repair proteins (e.g. H2AX, 53BP1, BRCA1, Rad51) following ionizing radiation? (This will support experimental data from both Figure 2 and Figure 7, as well as providing critical support for a role of this acetylation in the physiological DDR). Figure 7F should be backed up with the HR assay they used earlier.

10) In Figure 7E, overexpression of SIRT6 WT and K33R in SIRT1 KO cells can both rescue the genomic stability to a similar extent. Does it mean that only a small fraction of SIRT6 is acetylated in normal conditions? The authors should make some discussions.

11) A previous study demonstrated that PAR mediated recruitment of SIRT1 could also improve HR repair through deacetylating the chromatin remodeler BRG1 (PMID: 31291457). Also, SIRT6 activates PARP1 to promote HR and alt-NHEJ. In this study, SIRT6 was proved to be able to rescue deficient DNA repair caused by SIRT1 KO. The authors should comment on the regulatory loop

12) Please can the authors improve the clarity of their labelling on all figures and ensure sufficient information is provided in the figure legends.

13) In Figures 5F-G the authors describe experiments that suggest gH2AX is required for chromatin retention of SIRT6. Indeed in H2AX^-/-^ MEFs, SIRT6 is rapidly recruited to damage chromatin but is not retained at later time points. This is an interesting finding, although the quality of the IFs is questionable. Can the authors quantify these images? Since the experiment takes advantage of overexpressed H2AX proteins (which could cause artifacts), could the authors, as an alternative approach, also recapitulate this finding in WT MEFs in the presence of ATM/ATR inhibitors. ATM/ATR inhibition should reduce H2AX phosphorylation and therefore should mimic what is seen in Figure 5G.

14) In Figure 7A, an immunoblot for SIRT1 is required. In Figure 7B, an immunoblot is required showing the levels of expression of the SIRT6 constructs when re-expressed in the SIRT6 KO cells. This is critical for the interpretation of the comet assay. Indeed, it is somewhat surprising that the tail moment is not reduced to a greater extent when WT SIRT6 is re-introduced. Perhaps expression levels can explain this? This needs to be addressed by the authors.

15) In Figure 7C, can the authors mention the % of GFP expressing cells, rather than the relative levels? It is known that low-efficiency in GFP expression could lead to differences between samples that do not reflect true differences in HR.

16) In Figure 7D and E, the authors use an ectopic expression model in HeLa cells to test the ability of SIRT6 mutant expressing cells to form colonies in the absence and presence of ionizing radiation. This experiment needs to be repeated in the context of endogenous SIRT6 knockout background (like the SIRT6 HEK293KO cells they used in 7B). The reason for this is that the lack of difference seen with K33R could simply be that endogenous SIRT6 wild type levels are sufficient for colony formation and survival in these cells. Moreover, we absolutely need to see an immunoblot showing the expression levels of these mutants in these cells.

17) In Figure 7F the authors attempt to link the roles of SIRT1 and SIRT6 in the DNA damage response, suggesting that a SIRT6 mutant that cannot be acetylated at K33 is sufficient to overcome the effects of SIRT1 depletion. There is some difficulty in reconciling the result they got when re-expressing SIRT6 wild type in this experiment. If SIRT6 needs to be deacetylated in order to be recruited to DNA DSBs, then one would expect that overexpression of wild type SIRT6 in the absence of SIRT1 (SIRT1^-/-^ 293Ts) would have no impact on the ability of cells to repair DNA damage because it will still be acetylated at K33 and therefore not recruited to DNA DSBS. One would expect SIRT6 wild type to have comparable levels of DNA damage as the vector control, however the authors see a complete rescue of the phenotype when expressing WT SIRT6, raising concerns as to the significance of that acetylation. This needs to be explained/rationalized by the authors as this is somewhat contradictory to the model proposed by the authors. Again, we also need to see an immunoblot showing the expression levels of the SIRT6 mutants, as well as the successful re-expression of wild type SIRT1 into the SIRT1 ^-/-^ MEFS.

---

## [Author Response]

Essential revisions:1) In Figure 2C, the authors suggest that SIRT6 K33R does indeed have increased DSB affinity when compared to the K33Q mutant, and that this is supportive of the data presented in Figure 2B. Whilst it is true that K33R binds DNA more efficiently in this assay then K33Q, it is not possible to interpret the significance of the hypoacetylation for this binding without WT SIRT6. The authors need to repeat this experiment including recombinant WT SIRT6 as a control. One would expect that wild type SIRT6 DNA binding affinity should be somewhere in between the two mutants.

Thank you for the comment. As suggested, we repeated the experiment and included the data of WT SIRT6 in revised Figure 2C. The DNA binding affinity of WT SIRT6 was indeed in between SIRT6 K33R and SIRT6 K33Q.

2) In the text the authors state that the enzymatic activity of the SIRT6 K33Q is unaffected. However, based on the IF data in Figure 2 as well as the histone acetyl western blots in Figure 2—figure supplement 3, this does not seem to be the case. Can the authors please rationalize the differences seen in the IF and immunoblot experiments compared to their enzymatic activity assay described in Figure 2—figure supplement 3C.

Thank you for the question. The in vivo cellular data (revised Figures 2I and Figure 2—figure supplement 3A) showed that SIRT6 K33Q deacetylase activity towards histone H3 was compromised, while the in vitro deacetylation assay using the bacterially-purified SIRT6 proteins and synthesized acetyl H3 peptides showed negligible difference (revised Figure 2—figure supplement 3C). We reasoned that the initial recognition of DSB and retention on chromatin facilitated histone deacetylation mediated by SIRT6; the compromised DSB recognition and chromatin retention of K33Q inhibited SIRT6 deacetylase activity towards local histone, e.g. H3K9ac and H3K56ac. We have discussed this possibility in the revised manuscript.

3) For Figure 3G, the authors need to address the lack of pull down of the c-terminal peptide. The authors concluded that N terminus of SIRT6 is crucial for SIRT1-SIRT6 interaction. However, in Figure 3G, both SIRT6ΔN and ΔC fail to interact with SIRT1. Although the authors claimed that the result arises from the nuclear location signal is located in C terminus, a more direct evidence such as in vitro immunoprecipitation of SIRT6ΔC with SIRT1 is recommended.

Thank you for the suggestion. Indeed, we performed in vitro immunoprecipitation of GST-SIRT6 ΔC with His-SIRT1 in the presence or absence of a completing K33ac peptide. The data showed that GST-SIRT6 ΔC could pull down His-SIRT1 in the absence of K33ac peptide, while GST-SIRT6 ΔN failed (revised Figure 4H).

4) The biochemistry experiments performed in Figure 3 are nicely done. Do the authors see an increase in the interaction between SIRT6 and SIRT1 upon DNA damage (e.g. Ionizing Radiation)? This would be a strong experiment to include.

We appreciate the reviewers’ suggestion. We have conducted this experiment and the data showed that the interaction between SIRT6 and SIRT1 was increased upon DNA damage induced by CPT (revised Figure 7A).

5) Figure 4 needs some clarification with labeling and constructs being used. Labeling in 4A is unclear. In Figures 4C and 4D the authors ectopically express Flag-tagged SIRT1 and SIRT6 constructs. They then perform a Flag IP to isolate SIRT6 to determine the acetylation levels of SIRT6 in the presence of wild type SIRT1, enzymatically inactive SIRT1 or in the presence of inhibitors (Figure 4D). This makes interpretation of this data difficult, as it's not clear how the authors know which sirtuin is being immunoprecipitating when both are flag-tagged and the immunoprecipitation is being performed using a flag antibody. These experiments need to be repeated with a different tag on either SIRT1 or SIRT6 if this is the case. In Figure 4E, is the flag immunoblot the input or the IP? Going by the labelling, it appears as if this is the input. If this is the case, then there is no FLAG IP immunoblot shown and interpretation of the AcK immunoblot is inhibited without this data. Please provide the Flag IP immunoblot for this experiment.

We thank the reviewers for the comments and suggestions. We now clearly re-labeled the gel in revised Figure 4A. As suggested, we repeated the IP experiment with anti HA antibodies in cells expressing HA-labeled SIRT6 and FLAG-labeled SIRT1 WT or HY mutant, showing that while FLAG-SIRT1 WT deceased the acetylation level of HA-SIRT6, FLAG-SIRT1 HY failed both in cell system and in vitro (revised Figures 4C and 4D). Also, we provided the FLAG IP immunoblot in revised Figure 4F.

6) Explain the elevated γH2AX level in the absence of CPT in Input of Figure 5B.

We apologize for the confusion due to mislabeling. Indeed, we mixed cell lysate from CPT-treated and untreated samples and performed the IgG immunoprecipitation. We have included this information in revised Figures 5A and 5B.

7) Several gel images need to be repeated for clarity, as the conclusions drawn from them are critical to the thesis, but the ability to discern the accuracy of the gel images is weak. In particular, Figure 2G, 4D, 5E and 7A. Figure 7A, in particular, is very unclear (SIRT1 ^-/-^ bands look to be different sizes compared to the other lanes).

We appreciate the reviewers’ critical comments. We have repeated these experiments with high quality data (revised Figures 2G, 4D, 5E and 7B).

8) Many gel images and subsequent assays lack quantification. All images and laser experiments should include quantification to more accurately present the observations, as well as to provide the statistical rigor used. The graphics of Western blot need to have a uniform format (some with borders, some without). In regards to statistics, the authors need to clarify the n-values used for the vast majority of their experiments.

Thank you for the comments and suggestions. Now all images and laser experiments in the revised manuscript were quantified, analyzed with proper statistical methods, and n-values were also included. The graphics of western blot were reorganized in a uniform format.

9) Using the ISce1 system, the authors demonstrate that K33R is as effective as recognizing DNA DSBS as wild type SIRT6, and far superior to K33Q and HY mutants. One would therefore expect that the DNA repair kinetics of K33R is far superior to K33Q and HY mutants, and at least comparable to SIRT6 wild type. Can the authors show this by performing immunofluorescence staining/foci analysis for key DNA repair proteins (e.g. H2AX, 53BP1, BRCA1, Rad51) following ionizing radiation? (This will support experimental data from both Figure 2 and Figure 7, as well as providing critical support for a role of this acetylation in the physiological DDR). Figure 7F should be backed up with the HR assay they used earlier.

We appreciate the reviewers for the suggestion. We did immunofluorescence staining of γH2AX in Hela cells ectopically overexpressing SIRT6 WT and mutants (KR, KQ, HY) after IR. Less γH2AX foci was noticed in Hela cells transfected ectopic SIRT6 WT or K33R compared to K33Q or H133Y at 8h after IR (revised Figure 7—figure supplement 4), indicating that deacetylation of SIRT6 K33 is crucial for DNA repair. Also, we did the HR assay and the result confirmed that SIRT6 WT and K33R, but neither K33Q nor H133Y rescued the HR defect caused by *SIRT1* deficiency (revised Figures 7I and Figure 7—figure supplement 5B)

10) In Figure 7E, overexpression of SIRT6 WT and K33R in SIRT1 KO cells can both rescue the genomic stability to a similar extent. Does it mean that only a small fraction of SIRT6 is acetylated in normal conditions? The authors should make some discussions.

We appreciate the reviewers’ suggestion. In revised Figures 7G and 7H, SIRT6 WT and K33R had similar function in rescuing genomic stability in *SIRT1* KO cells. We reasoned that the overexpressed exogenous SIRT6 WT might not be effectively acetylated (revised Figure 4B). We have discussed this possibility in the revised manuscript.

11) A previous study demonstrated that PAR mediated recruitment of SIRT1 could also improve HR repair through deacetylating the chromatin remodeler BRG1 (PMID: 31291457). Also, SIRT6 activates PARP1 to promote HR and alt-NHEJ. In this study, SIRT6 was proved to be able to rescue deficient DNA repair caused by SIRT1 KO. The authors should comment on the regulatory loop

Thank you for the comment. A recent study demonstrated that PAR recruits SIRT1 and BRG1 to DSB sites and promotes HR efficiency (Chen et al., 2019). Other studies reported that SIRT6 mono-ADP ribosylates PARP1 to stimulate its activity (Mao et al., 2011) and is required for the recruitment of SNF2H to DSBs (Toiber et al., 2013). Both BRG1 and SNF2H are chromatin remodeling ATPases, responsible for opening the chromatin architecture. Our data revealed that both SIRT1 and SIRT6 mobilized to DSBs within seconds and SIRT1 deacetylates SIRT6 to promote its recognition of DSBs, highlighting their synergistic function in the early stage of DDR. Although the regulation loop is obscure, there might be one possibility that these early responding factors like PARP1, SIRT1, SIRT6, SNF2H and BRG1 are sequentially stimulated by DSBs, wherein they constitute a super complex to potentiate DDR and DNA repair; posttranslational modifications like deacetylation and mono-ADP ribosylation might empower the complex to recruit other repair factors more efficiently.

12) Please can the authors improve the clarity of their labelling on all figures and ensure sufficient information is provided in the figure legends.

As suggested, figure legends with more details were provided in revised manuscript.

13) In Figures 5F-G the authors describe experiments that suggest gH2AX is required for chromatin retention of SIRT6. Indeed in H2AX^-/-^ MEFs, SIRT6 is rapidly recruited to damage chromatin but is not retained at later time points. This is an interesting finding, although the quality of the IFs is questionable. Can the authors quantify these images? Since the experiment takes advantage of overexpressed H2AX proteins (which could cause artifacts), could the authors, as an alternative approach, also recapitulate this finding in WT MEFs in the presence of ATM/ATR inhibitors. ATM/ATR inhibition should reduce H2AX phosphorylation and therefore should mimic what is seen in Figure 5G.

We appreciate the comment. As suggested, the quantification of the images from laser damage experiments were provided in revised manuscript. Also, we used ATM/ATR inhibitor caffeine and performed the recruitment assay of SIRT6. The data showed that caffeine inhibited γH2AX. While the recruitment of SIRT6 to DSBs was merely affected at 2 min after laser-induced DNA damage, the retention of SIRT6 at DSBs was significantly compromised at 10 min after DNA damage in MEFs (revised Figure 5—figure supplement 1), supporting that γH2AX is required for SIRT6 retention surrounding DSBs.

14) In Figure 7A, an immunoblot for SIRT1 is required. In Figure 7B, an immunoblot is required showing the levels of expression of the SIRT6 constructs when re-expressed in the SIRT6 KO cells. This is critical for the interpretation of the comet assay. Indeed, it is somewhat surprising that the tail moment is not reduced to a greater extent when WT SIRT6 is re-introduced. Perhaps expression levels can explain this? This needs to be addressed by the authors.

We thank the reviewers for the suggestion. We repeated the experiment mentioned in revised Figure 7B and immunoblot was also provided for the comet assay in revised Figure 7—figure supplement 1A. Indeed, we noticed that the expression level of SIRT6 WT is lower than that of SIRT6 K33R, which might explain the milder reduction of tail moment after re-expression of SIRT6 WT than K33R in *SIRT6* KO cells (revised Figures 7D and 7E). Another possibility is that the re-expressed SIRT6 WT was partially acetylated in *SIRT6* KO cells whereas K33R was all “hypo-acetylated”.

15) In Figure 7C, can the authors mention the % of GFP expressing cells, rather than the relative levels? It is known that low-efficiency in GFP expression could lead to differences between samples that do not reflect true differences in HR.

We thank the reviewers for the suggestion. The percentage of GFP expressing cells was provided in revised Figure 7F.

16) In Figure 7D and E, the authors use an ectopic expression model in HeLa cells to test the ability of SIRT6 mutant expressing cells to form colonies in the absence and presence of ionizing radiation. This experiment needs to be repeated in the context of endogenous SIRT6 knockout background (like the SIRT6 HEK293KO cells they used in 7B). The reason for this is that the lack of difference seen with K33R could simply be that endogenous SIRT6 wild type levels are sufficient for colony formation and survival in these cells. Moreover, we absolutely need to see an immunoblot showing the expression levels of these mutants in these cells.

We thank the reviewers for the suggestion. Immunoblots of ectopic SIRT6 and SIRT1 expression levels were provided in revised Figure 7—figure supplement 1-3 and 5. As suggested, we tried many times to repeat the colony formation assay in *SIRT6* KO HEK293 cells. However, the cells failed to form colony after CPT or IR treatment owing to the poor colony-forming property of HEK293 cells. We then assessed cell viability using MTS assay in *SIRT6* KO HEK293 cells reconstituted with different SIRT6 mutants. The data showed that viability of cells transfected with SIRT6 WT or KR was much higher than those transfected with vector, SIRT6 KQ or HY after CPT or IR treatment (revised Figure 7—figure supplement 2). Also notably, the cell viability was lack of significant difference between groups after IR (2Gy) or CPT (1 μM) treatment in Hela cells, suggesting that endogenous SIRT6 is capable to maintain genomic stability upon low dose of DNA damage inducer (revised Figure 7—figure supplement 3).

17) In Figure 7F the authors attempt to link the roles of SIRT1 and SIRT6 in the DNA damage response, suggesting that a SIRT6 mutant that cannot be acetylated at K33 is sufficient to overcome the effects of SIRT1 depletion. There is some difficulty in reconciling the result they got when re-expressing SIRT6 wild type in this experiment. If SIRT6 needs to be deacetylated in order to be recruited to DNA DSBs, then one would expect that overexpression of wild type SIRT6 in the absence of SIRT1 (SIRT1^-/-^ 293Ts) would have no impact on the ability of cells to repair DNA damage because it will still be acetylated at K33 and therefore not recruited to DNA DSBS. One would expect SIRT6 wild type to have comparable levels of DNA damage as the vector control, however the authors see a complete rescue of the phenotype when expressing WT SIRT6, raising concerns as to the significance of that acetylation. This needs to be explained/rationalized by the authors as this is somewhat contradictory to the model proposed by the authors. Again, we also need to see an immunoblot showing the expression levels of the SIRT6 mutants, as well as the successful re-expression of wild type SIRT1 into the SIRT1 ^-/-^ MEFS.

We thank the reviewers for the comments. Please refer to our earlier response to point #10. The immunoblots of expression levels of the SIRT6 mutants and re-expression of wild type SIRT1 were provided in revised manuscript (revised Figure 7—figure supplement 5).